# 📈📊 Factuality Matters: When Image Generation and Editing Meet Structured Visuals

**Le Zhuo**[1,3*], **Songhao Han**[2*], **Yuandong Pu**[4,5], **Boxiang Qiu**[2], **Sayak Paul**[6],
**Yue Liao**[7], **Yihao Liu**[5], **Jie Shao**[8], **Xi Chen**[9], **Si Liu**[2†], **Hongsheng Li**[1†]

[1]CUHK MMLab, [2]Beihang University, [3]Krea AI, [4]Shanghai Jiao Tong University,
[5]Shanghai AI Lab, [6]Hugging Face, [7]National University of Singapore,
[8]ByteDance, [9]The University of Hong Kong

structvisuals.github.io

## Abstract

While modern visual generation models excel at creating aesthetically pleasing natural images, they struggle with producing or editing structured visuals like charts, diagrams, and mathematical figures, which demand composition planning, text rendering, and multimodal reasoning for factual fidelity. To address this, we present the first comprehensive, systematic investigation of this domain, encompassing data construction, model training, and an evaluation benchmark. First, we construct a large-scale dataset of 1.3 million high-quality structured image pairs derived from executable drawing programs and augmented with chain-of-thought reasoning annotations. Building on it, we train a unified model that integrates a VLM with FLUX.1 Kontext via a lightweight connector for enhanced multimodal understanding. A three-stage training curriculum enables progressive feature alignment, knowledge infusion, and reasoning-augmented generation, further boosted by an external reasoner at inference time. Finally, we introduce StructBench, a novel benchmark for generation and editing with over 1,700 challenging instances, and an accompanying evaluation metric, StructScore, which employs a multi-round Q&A protocol to assess fine-grained factual accuracy. Evaluations of 15 models reveal that even leading closed-source systems remain far from satisfactory. Our model attains strong editing performance, and inference-time reasoning yields consistent gains across diverse architectures. By releasing the dataset, model, and benchmark, we aim to advance unified multimodal foundations for structured visuals.

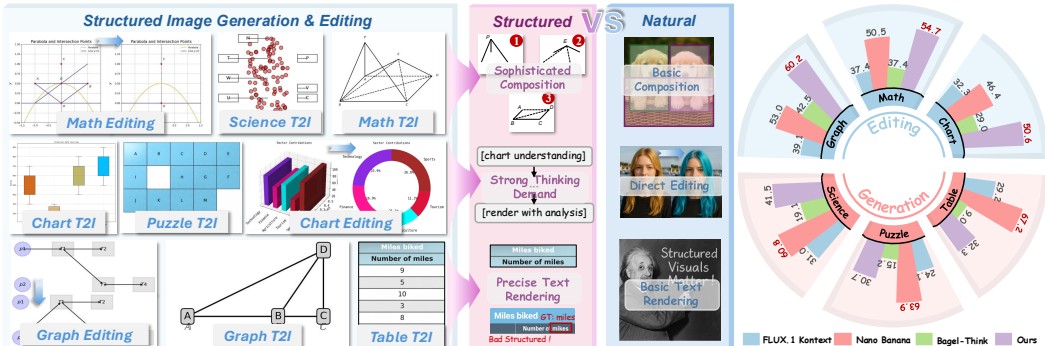

Figure 1: **Overview of our work.** *Left:* We showcase the diverse text-to-image (T2I) and editing examples from our dataset. In contrast to natural images, modeling structured visual demands sophisticated composition planning, strong multimodal understanding, and precise text rendering, as highlighted by the three key characteristics. *Right:* Our model demonstrates competitive performance against leading closed-source systems in both structured image generation and editing benchmarks.

---

*Equal Contribution.

†Corresponding Authors.

# 1 INTRODUCTION

Recent advances in visual generation have enabled models to follow user instructions and produce highly aesthetic images that are often indistinguishable from professional photography (Esser et al., 2024; Zhuo et al., 2024; Labs, 2024; Qin et al., 2025; Xie et al., 2025a; Wu et al., 2025a). With the emergence of unified multimodal models, such as GPT-Image (OpenAI, 2025b), Nano Banana (Google, 2025), and Bagel (Deng et al., 2025), these systems can better encode multimodal contexts and go beyond text-to-image generation to support more sophisticated visual tasks, including free-form image editing, style transfer, and more.

Despite producing increasingly appealing and intricate images, current state-of-the-art models struggle to accurately generate or edit charts, mathematical figures, and diagrams. We argue that modeling such structured visual components requires more than making images look aesthetically appealing; it demands accurate factuality, *i.e.*, encompassing capabilities like composition planning, precise text rendering, and multimodal reasoning as illustrated in Fig. 1. For instance, effective image editing in this domain necessitates robust understanding and extraction of multimodal representation from the input image and instruction. In contrast, vision-language models (VLMs) have made notable progress in understanding structured images (Fu et al., 2025; Wang et al., 2024; Zhang et al., 2024), revealing a pronounced gap between visual generation and understanding that hinders unified modeling. Consequently, structured visuals represent a crucial yet underexplored setting for multimodal generation. However, most existing works remain focused on aesthetics (Wu et al., 2023; Ma et al., 2025) or instruction following (Ghosh et al., 2024; Huang et al., 2023; Ye et al., 2025b) for natural images, leaving these unique challenges and their evaluation insufficiently addressed.

In this paper, we present the first systematic investigation into structured image generation and editing, including a comprehensive benchmark, a large-scale training corpus with chain-of-thought (CoT) annotations, and a strong unified model. Observing that many structured images map naturally to executable code, we collect millions of drawing programs and render them into seed images. Our pipeline performs edits at the code level to construct paired code-editing examples, which are then rendered to produce image-editing pairs. Unlike conventional image datasets (Sun et al., 2023; Wei et al., 2024; Ye et al., 2025b), images in our dataset are strictly aligned with their source codes, and editing instructions are driven by concrete code editing actions, yielding precise and verifiable state transitions. The final dataset comprises 1.3M high-quality image pairs with both text prompts and editing instructions, all annotated and filtered by GPT-5 (OpenAI, 2025a). Moreover, each sample is augmented with GPT-5-generated CoT reasoning annotations, further providing explicit reasoning trajectories for both structured image generation and editing.

Building on our dataset with rich annotations, we train a unified model for image generation and editing across both natural and non-natural domains based on FLUX.1 Kontext (Batifol et al., 2025). Different from approaches such as MetaQuery (Pan et al., 2025) that rely on heavy transformer-based projectors, we employ a lightweight MLP connector to align multimodal features from Qwen-VL (Bai et al., 2025) with FLUX.1 Kontext backbone, thereby improving the model's understanding of multimodal inputs such as charts and mathematical figures. We further design a three-stage training curriculum to progressively achieve feature alignment, knowledge infusion, and reasoning-augmented generation. After the final stage, we are able to decompose complex generation and editing tasks by leveraging an external reasoner to scale up inference-time compute for analysis and planning to enhance performance (Zhuo et al., 2025; Fang et al., 2025a).

Finally, we introduce **StructBench**, which contains over 1,700 high-quality samples carefully selected from six categories defined by structural characteristics. Unlike natural images, evaluating structured visuals is more challenging and demands reliable assessment of fine-grained details. To this end, we design a novel metric named **StructScore** that improves upon the naive "VLM-as-a-Judge" paradigm (Lin et al., 2024) and reduces hallucinations. Concretely, we utilize VLMs in a multi-round process to generate a set of fine-grained question-answer pairs that comprehensively probe all salient visual elements. We then elicit predicted answers from model-generated images and leverage VLMs to compare them against ground-truth responses, producing an overall score.

We comprehensively evaluate 15 models on our text-to-image and editing benchmarks, covering both open- and closed-source systems. Results reveal that, while closed-source models substantially outperform open-source ones, even state-of-the-art systems are far from satisfactory, consistent with our initial observations. Notably, our model achieves the best performance on image editing, enabled

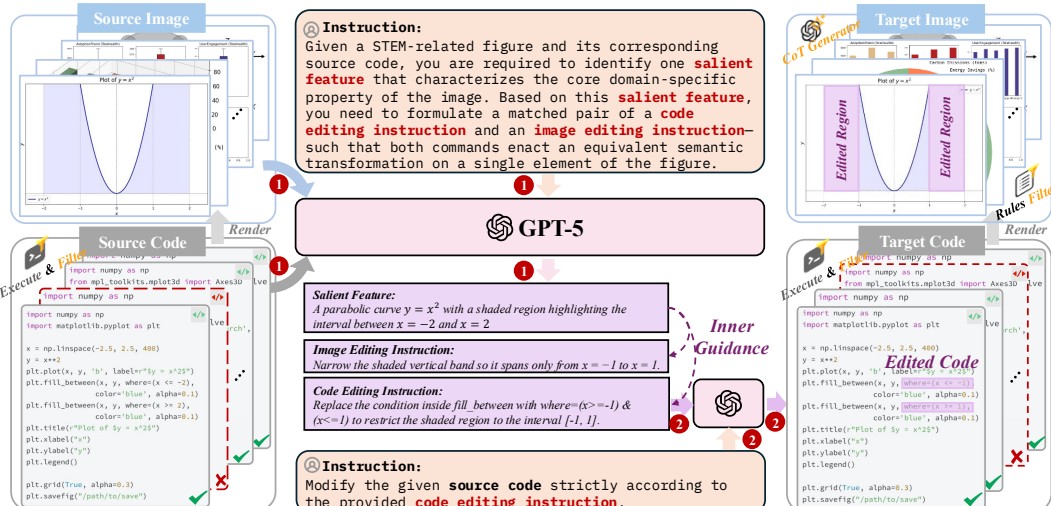

Figure 2: **Data construction pipeline.** We prompt GPT-5 to extract salient features, then generate paired editing instructions from the source code and rendered image. The source code is modified according to the code-editing instructions. The target image rendered from modified code is passed through rule-based filters to ensure the overall quality of the constructed dataset.

by adding explicit reasoning trajectories at inference time. We validate that applying the same reasoning procedure to other models yields consistent improvements, indicating that structured image generation and editing particularly benefit from scaling inference-time compute for reasoning, which is the key bottleneck in current unified systems. By releasing our dataset, model, and benchmark, we aim to draw broader attention in the multimodal community to structured visuals and accelerate progress toward truly unified foundation models.

**Benchmarks.** Existing T2I benchmarks (Wu et al., 2023; Ghosh et al., 2024; Huang et al., 2023) primarily emphasize visual quality and instruction following, with a particular focus on compositionality. Similarly, editing benchmarks (Sheynin et al., 2024; Wang et al., 2023; Ma et al., 2024; Ye et al., 2025b) center on visual consistency and adherence to editing instructions, frequently relying on similarity metrics such as DINO (Oquab et al., 2023) or CLIP (Radford et al., 2021) scores. As model capabilities have improved, recent benchmarks (Wu et al., 2025c; Zhao et al., 2025a; Sun et al., 2025; Cao et al., 2025) have begun to probe more challenging tasks that require world knowledge and reasoning for generation or editing. Since these evaluations mostly target natural images, their assessment protocols tend to be coarse-grained, using learnable scoring models or naive VLM-as-a-judge (Chai et al., 2024; Han et al., 2025b) pipelines, which are ill-suited for non-natural, structured imagery.

## 2 STRUCTURED IMAGE DATASET

### 2.1 DATA CURATION PIPELINE

A key challenge to training and evaluating models on structured images is the absence of open-source datasets tailored for generative modeling. Although the visual understanding community offers related resources (Masry et al., 2022; Zhang et al., 2024; Wang et al., 2024), their quality is inconsistent and they rarely support constructing precise image-editing pairs. Reviewing data curation pipelines for natural-image generation and editing, prior work (Sun et al., 2023; Wei et al., 2024; Fang et al., 2025b) typically collects user or synthetic prompts as seed instructions and then employs expert models as renderers to synthesize images. Given that structured images can be programmatically specified (Fu et al., 2025), a natural alternative is to collect source code as seed prompts, create code-editing operations to obtain target code, and leverage code-based graphics libraries as the renderer to automatically synthesize high-fidelity structured images and editing pairs.

**Code-Aligned Image Synthesis.** Concretely, we collect approximately 2M programs for rendering structured images from diverse sources (Belouadi et al., 2023; Wang et al., 2025b; Zhao et al., 2025b),

covering mathematics, charts, puzzles, scientific figures, graph diagrams, and tables, primarily in Python and LaTeX. We execute each program and retain only those that render a valid image, yielding source code–image pairs. The next step is to generate code-editing instructions to obtain the corresponding target code–image pairs. However, we observe that directly editing the code often yields overly specific, low-level actions that are not easily discernible at the visual level, whereas image editing is largely guided by perceptible visual elements. As illustrated in Fig. 2, we design a multi-step annotation process. Given the source code–image pair, GPT-5 (OpenAI, 2025a) first analyzes the visual characteristics of the source image, producing salient features (caption). Guided by these features, it then generates a matched pair of image-editing and code-editing instructions in parallel. This design ensures that the image-editing instruction is constrained to reference only visual elements, while the code-editing instruction specifies the precise program-level changes. Finally, GPT-5 applies the code-editing action to the source program to obtain the target code, which we execute to render the target image, yielding a strictly aligned and verifiable state transition.

## 2.2 POST-PROCESSING PIPELINE

**Filtering.** To ensure data quality and annotation richness, we implement a comprehensive post-processing pipeline. First, we discard invalid samples by executing each program and removing those that fail to render or produce corrupted outputs. Second, we apply heuristic, rule-based filters to eliminate (1) source–target pairs with null edits, *i.e.*, low perceptible visual difference, and (2) low-information images with minimal semantic content (*e.g.*, near-uniform or trivially simple renderings). The final dataset contains 1.3M examples spanning diverse categories. Each example comprises a source–target image pair, a caption of the source image for text-to-image generation, and an image editing instruction. Fig. 4(a) visualizes the percentage and number of each editing category.

**Reasoning Trajectory Synthesis.** Instructions in prior image generation and editing datasets are often overly terse (*e.g.*, "add tree right," "a photo of a soap bar"), offering insufficient semantic guidance. This limitation is especially problematic for structured visuals, where one-to-many mappings between instructions and outputs hinder both learning and evaluation. A distinct feature of our dataset is the inclusion of chain-of-thought (CoT) reasoning annotations. Specifically, each text-to-image example is paired with a carefully constructed dense caption with detailed attribute analysis, and each image-editing example is accompanied by a three-step reasoning chain (*i.e.*, input image analysis, editing instruction interpretation, target image prediction), all generated by GPT-5. As illustrated in Fig. 4(b,c), these annotations are substantially longer, providing rich and accurate semantic and analytical signals that better support complex structured image generation and editing.

## 3 STRUCTBENCH

### 3.1 DATA SELECTION

The dataset introduced in Sec. 2 is generated via code actions and filtered for validity, yielding high-quality samples that are well-suited for benchmark construction. To ensure diversity and balance, we first cluster images by salient visual features and then carefully curate six primary domains, including Math, Graph, Chart, Puzzle, Science, and Table, through manual taxonomy and selection. We apply stratified sampling within each domain, and all samples undergo dual review by GPT-5 and human annotators to verify image quality and instruction correctness.

### 3.2 EVALUATION PROTOCOL

Evaluating structured images differs from natural images in three fundamental respects: (1) structured images demand strict prompt following at both the global layout level and in fine-grained structural elements such as numerical values, dense text, and geometric primitives, whereas prompts for natural images are often short and permissive; (2) aesthetic criteria for structured images are less important and substantial different from those for natural images; and (3) structured-image domains exhibit pronounced inter-domain differences that call for adaptive, domain-specific evaluation criteria. Consequently, common metrics used in text-to-image or editing benchmarks, such as CLIP score, aesthetic score (Wu et al., 2023), and naive VLM-based scores (Lin et al., 2024), are not well-suited for this setting. To bridge this gap, we propose **StructScore**, a metric that leverages a VLM in a controlled multi-turn workflow to assess fine-grained, instance-specific attributes.

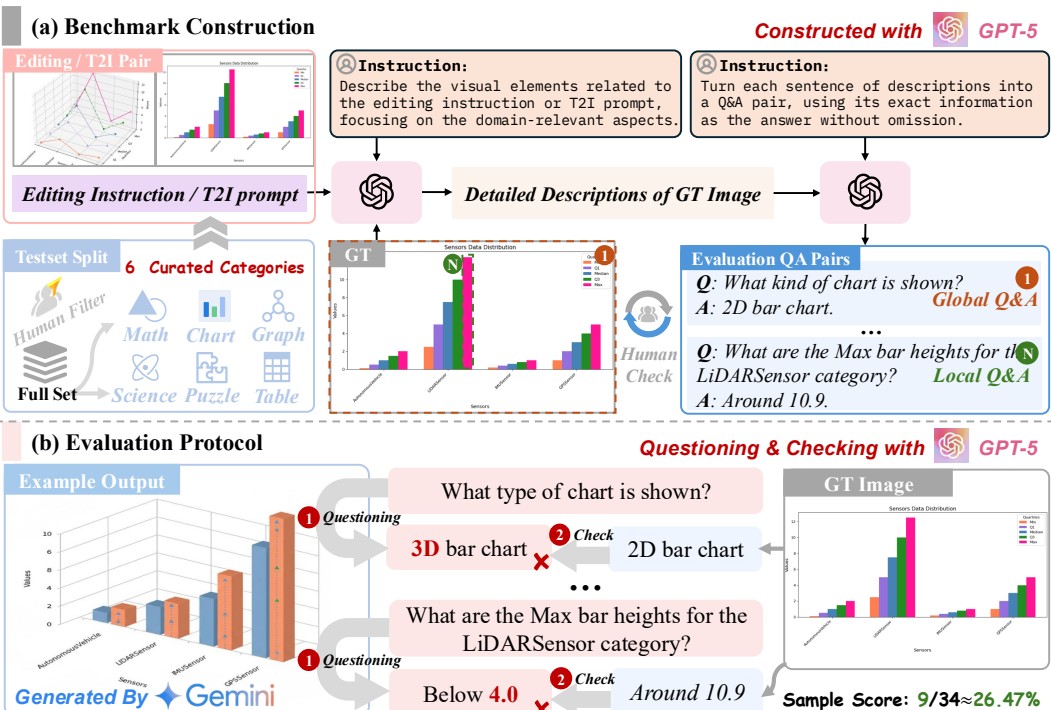

Figure 3: **Benchmark construction and evaluation workflow.** (a) Benchmark construction: We cluster the data into six categories, and for each editing and text-to-image (T2I) example, GPT-5 generates detailed image descriptions that are transformed into Q&A pairs for evaluating diverse visual aspects. (b) Evaluation protocol: Using the Q&A pairs, GPT-5 is queried on generated images for open-ended responses, which are compared with ground-truth answers to yield a final score.

**Q&A Construction.** Our goal is to build, for each data item, a set of Q&A pairs that can comprehensively assess generated or edited samples. Starting from the instruction and ground-truth image, we first prompt GPT-5 to describe all salient and relevant visual elements with their attributes. GPT-5 then decomposes this detailed description into sentences and converts each into a verification Q&A pair targeting specific attributes or relations, as illustrated in Fig. 3(a).

**Evaluation Metric.** A naive evaluation would binarize these Q&As and query a VLM on the model-generated image, yielding yes/no answers. However, this yields an unreliable ceiling as random guessing can achieve 50% accuracy. Instead, because our questions are atomic, we prompt the VLM to produce open-ended answers for each question based on the generated image, forming *[question, predicted-answer, ground-truth]* triplets. We then compare the predicted and ground-truth answers and average the accuracy of similarity scores across all Q&As to obtain the sample-level accuracy.

Unlike text-to-image generation, image editing cannot be fairly evaluated by simply averaging all Q&A scores. This is because editing simultaneously assesses visual consistency and instruction following, and visual consistency is substantially easier: a model can attain high scores via near-identity mappings. To disentangle these dimensions, we compute Q&A accuracy on both the source and ground-truth target images. If a Q&A is correct for both, it is labeled as visual-consistency related; if it is incorrect on the source but correct on the target, it is labeled as instruction-following related. This binary categorization reveals that visual-consistency Q&As are far more frequent than instruction-following Q&As, as the latter typically focus on small but critical changes. To counter this imbalance and prioritize instruction adherence, we report the final editing accuracy as a weighted score, *i.e.* $0.1\times$ visual-consistency accuracy and $0.9\times$ instruction-following accuracy. Appendix A.5 analyzes alternative weightings and their alignment with human preferences, where our chosen setting achieves the best correlation with human evaluations.

**Atomic Q&A Refinement Improves Metric Reliability.** We conduct strict human audits to regulate the number of Q&As per sample, validate their reliability, and ensure our metric alignment with

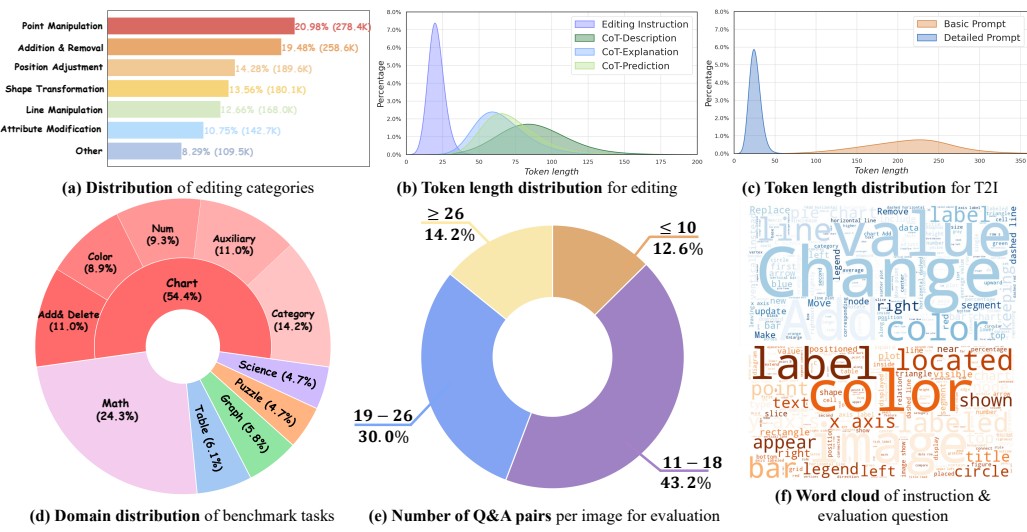

Figure 4: **Statistical analysis** of our dataset (a-c) and benchmark (d-f).

human judgments. As shown in Fig. 15, we observe that certain Q&A pairs, while appearing plausible to humans, are not clearly verifiable by the VLM evaluator. We attribute this issue to non-atomic questions that bundle multiple attributes or relations, thereby introducing answer stochasticity. To mitigate hallucination and enhance determinism, we revise the VLM prompts to split compound descriptions into multiple atomic items and enforce minimal, concise ground-truth answers. This refinement process nearly doubles the number of Q&A pairs, reflecting increased atomicity. A side-by-side comparison of the original and revised atomic Q&A pairs is provided in Fig. 15.

Given that our benchmark contains ground-truth images, we leverage the StructScore on these images as a proxy for metric reliability. An ideal metric should achieve 100% accuracy on ground-truth images; however, the initial StructScore is approximately 80%. Therefore, we prompt GPT-5 to rewrite the failing Q&A pairs, re-aligning them with ground-truth images to make them clear and unambiguous. Following this refinement, the updated Q&A set yields over 95% accuracy on ground-truth images, demonstrating substantially improved reliability.

## 3.3 NUMERICAL STATISTICS

Our StructBench consists of 1,714 items with 32,031 and 37,941 Q&A pairs for editing and generation respectively. Fig. 4(d) summarizes the distribution of StructBench across six domains. Given the importance of charts, we further stratify the chart test set into five categories by editing type. Fig. 4(e) reports the number of Q&A pairs per sample, where over 87% of samples contain more than ten questions, and 14% include more than twenty-six. Finally, the word clouds for editing instructions and questions in Fig. 4(e) highlight a focus on the key elements and attributes of structured images.

## 4 MODEL TRAINING

### 4.1 MODEL ARCHITECTURE

We build on FLUX.1 Kontext (Batifol et al., 2025), a diffusion transformer with unified image generation and editing capabilities, and adapt it to structured visual content. Following its original configuration, we encode textual instructions with T5 (Raffel et al., 2020)[1], and encode both the input image and the target image with a VAE. The resulting token streams are concatenated into a unified sequence and processed with joint attention. While the VAE supplies low-level features of the input image, structured image editing often depends on higher-level semantics. For example, converting a bar plot into a pie chart requires recognizing the underlying quantitative relationships to render the correct proportions. To enhance semantic perception, we utilize a lightweight MLP connector

---

[1] We discard CLIP features due to its limited token budget and observed no degradation in output quality.

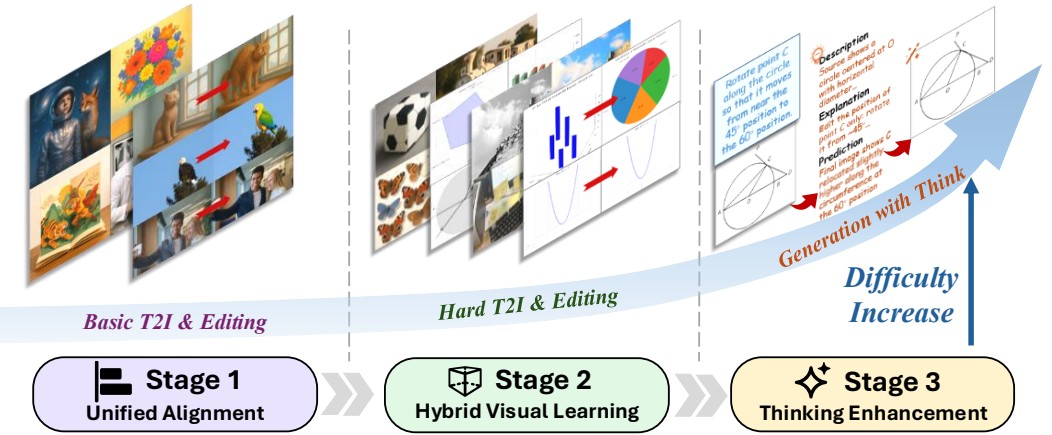

Figure 5: **The three-stage progressive training pipeline.** Training difficulty increases across stages, from alignment to hybrid visual learning and thinking enhancement.

that aligns Qwen-VL (Bai et al., 2025) encoded multimodal features with FLUX.1 Kontext. This deisgn choice reduces training overhead and, empirically, stabilizes optimization without sacrificing performance compared to learnable-query, transformer-based connectors (Pan et al., 2025).

## 4.2 TRAINING PIPELINE

To gradually introduce structured-visual knowledge to our model while maintaining its general capabilities, we adopt a three-stage progressive training pipeline, as illustrated in Fig. 5.

**Stage 1: Unified Alignment.** The goal of this stage is to align Qwen-VL's features with the diffusion backbone rather than injecting new knowledge. Accordingly, we freeze the backbone model and train only the newly added connector, using relatively simple text-to-image and editing data. To prevent the already well-aligned T5 encoder from becoming a shortcut that impedes connector alignment (Lin et al., 2025), we remove T5 features during this stage and rely solely on Qwen-VL features. After alignment training, the model can roughly follow instructions for both generation and editing.

**Stage 2: Hybrid Visual Learning.** We then incorporate our constructed structured text-to-image and editing datasets and jointly fine-tune the diffusion backbone and connector to inject domain knowledge of structured visuals. To preserve general-domain capabilities, we include a mixture of high-quality text-to-image and editing datasets. However, structured visuals exhibit markedly different pixel statistics from natural images, which contain large regions of uniform backgrounds and, in editing, small regions of change. Therefore, we introduce a mask-based training strategy that adaptively downweights losses on background and unchanged regions during training. Leveraging Qwen-VL's multimodal representations and the pretrained T5 encoder, the model improves on both structured and general domains.

**Stage 3: Thinking Enhancement.** We observe that our Stage-2 model, as well as other baselines, often produce images that appear visually plausible but contain semantic errors on complex tasks. We attribute this to an overreliance on shallow visual patterns driven by text features, coupled with insufficient grounding in the input image or prompt (e.g., layout or quantitative relationships). Because such semantic comprehension is a core strength of VLMs, we leverage the CoT annotations from Sec. 2.2 and open-source datasets (Fang et al., 2025b) as long-context inputs to Qwen-VL to inject explicit reasoning. This endows the model with the ability to follow complex multimodal reasoning instructions. It also enables scaling inference-time compute: a VLM first analyzes the input image–text pair and predicts the ideal target content, and the resulting expanded analysis is then fed to the generator to guide synthesis.

## 5 EXPERIMENTS

### 5.1 EXPERIMENTAL SETUP

**Training Details** We provide comprehensive training details in Appendix A.2, including model configurations, stage-specific hyperparameters, and the datasets used. Our base model is FLUX.1

Table 1: **Quantitative comparison on StructEditBench.** The table showcases the results of Accuracy (%) and PSNR (with ground-truth target image) for various methods. ■ ■ indicate the first and second Accuracy results, respectively.

| Model | Math Acc ↑ | Math PSNR ↑ | Chart Acc ↑ | Chart PSNR ↑ | Graph Acc ↑ | Graph PSNR ↑ | Puzzle Acc ↑ | Puzzle PSNR ↑ | Science Acc ↑ | Science PSNR ↑ | Table Acc ↑ | Table PSNR ↑ | Overall Acc ↑ | Overall PSNR ↑ |
|---|---|---|---|---|---|---|---|---|---|---|---|---|---|---|
| Nano Banana | 50.46 | 20.77 | 46.42 | 14.93 | 52.97 | 21.22 | 66.56 | 22.92 | 69.16 | 22.61 | 75.71 | 19.75 | 51.57 | 21.09 |
| GPT-Image | 51.49 | 17.06 | 45.82 | 12.56 | 50.71 | 17.24 | 76.03 | 16.55 | 67.61 | 16.61 | 83.26 | 14.35 | 52.2 | 16.64 |
| Seedream 4.0 | 51.06 | 23.63 | 46.83 | 16.54 | 51.72 | 24.12 | 71.13 | 26.93 | 69.22 | 26.46 | 88.19 | 24.75 | 52.85 | 24.45 |
| Nano Banana 2.0 | 64.64 | 24.88 | 63.28 | 16.74 | 72.58 | 26.39 | 81.07 | 28.12 | 71.19 | 26.71 | 91.10 | 25.14 | 67.05 | 25.59 |
| UniWorld-V1 | 9.41 | 8.84 | 5.99 | 7.87 | 8.83 | 6.16 | 9.11 | 7.71 | 19.91 | 7.67 | 16.13 | 8.24 | 8.40 | 8.21 |
| DiMOO | 26.79 | 21.56 | 16.52 | 14.98 | 24.03 | 21.77 | 29.52 | 22.26 | 26.08 | 22.57 | 24.64 | 19.47 | 21.00 | 21.49 |
| OmniGen2 | 29.44 | 15.95 | 18.55 | 12.44 | 34.63 | 11.31 | 28.61 | 16.51 | 39.55 | 15.60 | 30.36 | 16.60 | 24.30 | 15.49 |
| Ovis-U1 | 31.64 | 18.45 | 21.94 | 13.30 | 38.03 | 19.01 | 42.08 | 17.92 | 44.52 | 18.68 | 35.58 | 16.62 | 28.06 | 18.25 |
| HiDream-E1.1 | 28.07 | 18.43 | 26.36 | 12.91 | 29.63 | 18.26 | 43.77 | 18.04 | 36.66 | 16.47 | 48.79 | 17.12 | 29.63 | 18.01 |
| Bagel | 21.27 | 21.38 | 27.11 | 16.38 | 29.94 | 22.70 | 41.59 | 24.22 | 47.16 | 23.56 | 47.35 | 21.54 | 28.87 | 22.06 |
| Bagel-Think | 37.40 | 23.97 | 28.98 | 16.82 | 42.51 | 26.49 | 36.11 | 26.75 | 43.15 | 25.57 | 40.46 | 23.83 | 33.34 | 24.70 |
| Step1X-Edit | 34.47 | 23.41 | 28.05 | 16.68 | 33.26 | 24.56 | 60.48 | 25.94 | 46.47 | 24.98 | 57.81 | 23.97 | 34.11 | 24.03 |
| FLUX.1 Kontext | 37.36 | 19.78 | 32.29 | 14.61 | 39.12 | 20.10 | 58.35 | 20.38 | 55.99 | 20.99 | 58.05 | 18.52 | 37.56 | 19.84 |
| Qwen-Edit | 40.48 | 23.73 | 30.17 | 12.33 | 44.83 | 26.11 | 53.74 | 27.31 | 55.99 | 25.53 | 67.76 | 25.71 | 38.12 | 24.81 |
| Ours | 54.74 | 23.31 | 50.58 | 15.33 | 60.18 | 24.65 | 73.0 | 26.33 | 75.05 | 25.80 | 77.08 | 23.19 | 55.98 | 24.01 |

Table 2: **Quantitative comparison on StructT2IBench**, reporting Accuracy (%).

| Model | Chart | Graph | Math | Puzzle | Science | Table | Overall |
|---|---|---|---|---|---|---|---|
| Seedream 4.0 | 35.79 | 54.08 | 63.33 | 50.89 | 62.59 | 68.94 | 47.52 |
| Nano Banana | 35.55 | 58.96 | 64.81 | 63.87 | 60.75 | 67.20 | 48.45 |
| GPT-Image | 37.09 | 57.00 | 63.25 | 59.42 | 60.94 | 83.31 | 49.58 |
| Nano Banana 2.0 | 93.79 | 90.51 | 89.04 | 88.42 | 87.69 | 95.35 | 92.00 |
| UniWorld-V1 | 1.71 | 5.52 | 4.72 | 1.58 | 8.82 | 5.25 | 3.20 |
| Bagel | 4.66 | 3.61 | 4.02 | 4.46 | 8.60 | 5.74 | 4.69 |
| Bagel-Think | 4.81 | 15.33 | 13.89 | 15.22 | 19.05 | 8.97 | 9.03 |
| HiDream-I1-Full | 9.47 | 20.84 | 19.20 | 18.00 | 26.77 | 27.05 | 14.77 |
| OmniGen2 | 10.67 | 22.51 | 22.89 | 18.63 | 28.00 | 22.61 | 16.24 |
| FLUX.1 Dev | 12.35 | 20.09 | 19.86 | 20.63 | 25.25 | 27.00 | 16.51 |
| FLUX.1 Kontext | 17.22 | 24.64 | 21.42 | 24.06 | 30.97 | 29.16 | 20.36 |
| Ovis-U1 | 24.75 | 16.08 | 19.45 | 21.23 | 26.03 | 12.70 | 22.83 |
| Qwen-Image | 32.23 | 48.05 | 46.98 | 48.90 | 53.51 | 73.65 | 41.03 |
| Ours | 20.91 | 33.45 | 41.70 | 30.66 | 41.46 | 32.26 | 28.80 |

Kontext [dev] (Batifol et al., 2025), where we remove the original CLIP encoder and replace it with Qwen2.5-VL-7B (Bai et al., 2025) to encode both the input instructions and images. Throughout training, we adopt dynamic-resolution sampling, where images are resized to lie near 512×512 while preserving their native aspect ratios, maximizing information retention. During inference, we employ GPT-5 (OpenAI, 2025a) as an external reasoner to provide reasoning trajectories from inputs.

**Evaluation Details** We evaluate 15 closed- and open-source models, covering the most recent text-to-image, image-editing, and unified models. Closed-source systems include GPT-Image (OpenAI, 2025b), Nano Banana (Google, 2025), and Seedream 4.0 (Seedream et al., 2025). Open-source baselines include FLUX.1-dev (Labs, 2024), FLUX.1 Kontext (Batifol et al., 2025), Step1X-Edit (Liu et al., 2025), Bagel (Deng et al., 2025), Bagel-Think (Deng et al., 2025), HiDream-E1.1 (Cai et al., 2025), UniWorld-V1 (Lin et al., 2025), Ovis-U1 (Wang et al., 2025a), OmniGen2 (Wu et al., 2025b), Qwen-Image (Wu et al., 2025a), and DiMOO (Team, 2025). For aspect ratios, models that support custom output sizes are configured to match the ground-truth image proportions. To ensure fairness and reproducibility, all models are run with the default settings from their official repositories and generated on H200 GPUs. For the LLM-based evaluator, we report results using both the current state-of-the-art closed-source LLM (GPT-5 (OpenAI, 2025a)) in the main text and the leading open-source alternative (Qwen2.5-VL-72B (Bai et al., 2025)) in Appendix A.3. We also release the complete prompts used for data construction and evaluation in Appendix A.7.

## 5.2 Results and Analysis

We report the performance of 15 models on 6 subtasks of StructEditBench and StructT2IBench in Tab. 1 and Tab. 2, respectively. To provide finer-grained analysis, we break down chart-related editing by editing type in Tab. 3. Case study demos are provided in Appendix A.6. The main findings are:

Table 3: **Quantitative comparison on StructEditBench (charts only).** ▨ ▨ indicate the first and second Accuracy results, respectively.

| Model | Category | | Color | | Num | | Auxiliary | | Add&Del | | Overall | |
|---|---|---|---|---|---|---|---|---|---|---|---|---|
| | Acc ↑ | PSNR ↑ | Acc ↑ | PSNR ↑ | Acc ↑ | PSNR ↑ | Acc ↑ | PSNR ↑ | Acc ↑ | PSNR ↑ | Acc ↑ | PSNR ↑ |
| GPT-Image | 40.62 | 9.85 | 54.57 | 13.91 | 33.12 | 13.48 | 64.03 | 13.48 | 38.02 | 12.73 | 45.82 | 12.56 |
| Nano Banana | 39.75 | 10.33 | 54.34 | 17.59 | 35.64 | 16.56 | 67.40 | 17.39 | 36.77 | 13.88 | 46.42 | 14.93 |
| Seedream 4.0 | 38.13 | 9.67 | 61.84 | 21.77 | 36.00 | 19.22 | 65.92 | 19.46 | 36.04 | 14.28 | 46.83 | 16.54 |
| Nano Banana 2.0 | **53.91** | 10.38 | **69.21** | 21.22 | **60.11** | 19.30 | **73.53** | 19.09 | **63.04** | 15.08 | **63.28** | 16.74 |
| UniWorld-V1 | 5.98 | 7.60 | 8.19 | 8.29 | 2.58 | 7.57 | 10.24 | 8.62 | 2.81 | 7.35 | 5.99 | 7.87 |
| DiMOO | 11.20 | 9.82 | 15.31 | 16.97 | 17.39 | 17.30 | 21.57 | 17.59 | 18.57 | 14.46 | 16.52 | 14.98 |
| OmniGen2 | 17.30 | 8.61 | 28.84 | 11.78 | 15.48 | 14.03 | 22.42 | 16.59 | 10.54 | 12.11 | 18.55 | 12.44 |
| Ovis-U1 | 18.15 | 9.57 | 30.68 | 15.38 | 20.79 | 15.13 | 25.49 | 14.25 | 17.21 | 13.11 | 21.94 | 13.30 |
| HiDream-E1.1 | 22.69 | 9.29 | 39.86 | 14.07 | 21.49 | 15.04 | 32.65 | 14.34 | 18.05 | 12.71 | 26.36 | 12.91 |
| Bagel | 25.69 | 9.08 | 38.20 | 20.46 | 26.30 | 20.29 | 30.00 | 21.24 | 17.79 | 14.93 | 27.11 | 16.82 |
| Step1X-Edit | 21.96 | 10.40 | 36.51 | 19.75 | 25.46 | 20.22 | 34.40 | 19.46 | 24.92 | 15.11 | 28.05 | 16.68 |
| Bagel-Think | 24.55 | 9.00 | 45.46 | 19.35 | 25.60 | 20.14 | 34.54 | 20.62 | 18.69 | 14.58 | 28.98 | 16.38 |
| Qwen-Edit | 23.53 | 9.52 | 41.80 | 13.63 | 22.90 | 13.69 | 42.39 | 13.07 | 23.27 | 12.46 | 30.17 | 12.33 |
| FLUX.1 Kontext | 24.67 | 10.14 | 44.56 | 16.53 | 29.24 | 16.74 | 44.02 | 16.79 | 23.06 | 13.95 | 32.29 | 14.61 |
| Ours | 50.81 | 10.40 | 64.10 | 18.16 | 33.45 | 17.04 | 66.34 | 17.90 | 38.10 | 14.37 | 50.58 | 15.33 |

**Closed-source models lead, yet all models remain far from satisfactory.** Across both editing and T2I benchmarks, closed-source systems consistently occupy the top positions. For example, on T2I, the best closed-source model (GPT-Image) attains a score of 49.58, whereas the strongest open-source baseline (Qwen-Image) achieves 41.03. Notably, our model achieves the highest score (55.98) on the StructEditBench. However, even the best model achieves only about half accuracy on either generation or editing, highlighting substantial room for improvement on structured visuals.

**Data, rather than architecture, is the dominant driver of performance.** Among open-source models, varying visual encoders (e.g., VAE (Batifol et al., 2025), Qwen-VL (Bai et al., 2025), SigLIP (Zhai et al., 2023)) and modeling paradigms (diffusion transformer, unified autoregressive transformer, discrete diffusion) do not yield a consistent winner. By training on our curated dataset, our model delivers strong performance gains over the original FLUX.1 Kontext, underscoring the importance of data scale and quality. Instead, models trained predominantly on natural-image corpora with limited task coverage (e.g., UniWorld-V1) significantly lag behind on structured-image tasks. Besides, the performance gap is larger for T2I than for editing, likely because editing emphasizes transferable operations like add, delete, move, whereas T2I requires synthesizing fine-grained attributes from scratch, a substantially more challenging objective.

**Reasoning capability has a substantial impact.** For relatively simple edits such as modifying colors or adding auxiliary lines (Tab. 3), most models approach 50% accuracy. In contrast, performance drops sharply on more complex tasks like chart-type conversion, which require first understanding the input (e.g., proportions and ordering) and then generating the target according to structural specifications. Moreover, both the "thinking" variant Bagel-Think and our model with explicit reasoning substantially outperform non-reasoning baselines in both generation and editing, underscoring the benefits of incorporating reasoning before generation.

## 5.3 SCALING COMPUTE WITH EXPLICIT REASONING

Motivated by the strong performance of our model, we further investigate the impact of adding explicit test-time reasoning for structured image editing. We introduce an external reasoner (GPT-5) during inference that performs a three-step analysis over the instruction and input image: (1) summarize salient visual elements; (2) localize the elements to be edited and specify the intended changes; and (3) forecast the post-editing outcome. This design decouples complex visual reasoning and planning from image synthesis, allowing the generator to focus on the generation task itself.

As shown in Fig. 7 and 8, providing explicit reasoning trajectories yields substantial improvements for most models. Notably, Bagel augmented with our carefully designed reasoning trajectories significantly outperforms its native "thinking" variant, Bagel-Think (38.44 vs. 33.34), indicating that the trajectory design and quality are critical. Moreover, the family of unified models such as GPT-Image and Bagel benefits more from added reasoning than specialized expert models like

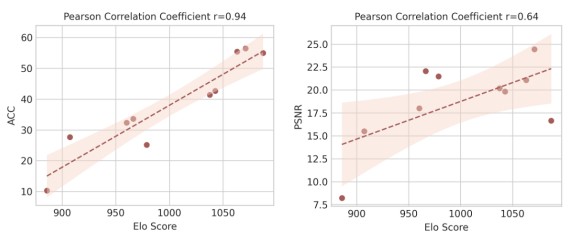 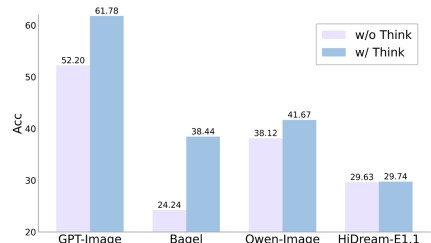

Figure 6: **Pearson correlation analysis** among Accuracy, PSNR, and human Elo score for editing models.

Figure 7: **Study of explicit reasoning** added to different models.

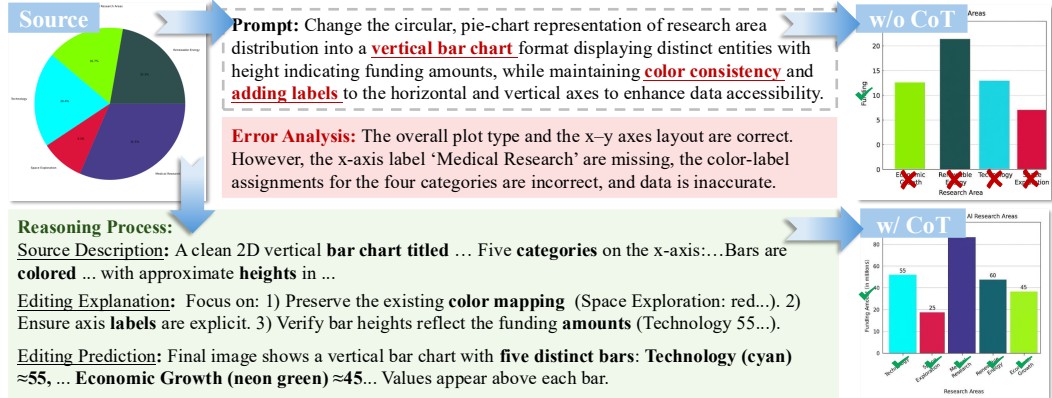

Figure 8: **Case study: explicit reasoning improves fidelity.** When dealing with complex tasks, direct generation often leads to plausible yet incorrect images. Incorporating a reasoner to explicitly describe, analyze, and predict enables the generator to produce faithful, correct outputs.

HiDream-E1.1. Overall, these results again validate that high-quality multimodal reasoning is essential for complex editing. Whether via interleaved multimodal pretraining or reasoning-centric post-training, unified models with native multimodal reasoning appear to be a promising path forward.

## 5.4 HUMAN ALIGNMENT STUDY

To evaluate how well StructScore aligns with human judgments, we conduct a large-scale Elo rating study to elicit human preference rankings. Implementation details and full results are provided in Appendix A.4. We then compute the Pearson correlation coefficient ($r$) between the human Elo rankings and scores from various metrics across multiple benchmarks. As shown in Fig. 6, StructScore achieves a strong correlation with human preferences ($r > 0.9$), substantially exceeding traditional metrics like PSNR. This verifies the effectiveness of breaking down complex evaluations into atomic questions and indicates that StructScore is a reliable proxy for human assessment.

## 6 CONCLUSION

In this work, we aim to address the critical challenge of structured image generation and editing, a domain where existing models fall short due to the high demand for factual accuracy. We introduced a comprehensive solution, including a large-scale, code-aligned dataset with chain-of-thought annotations, a strong unified model trained with a progressive curriculum, and a rigorous benchmark-metric suite for fine-grained, low-hallucination evaluation. Our experiments demonstrate that even leading models struggle with structured visuals, yet our approach sets a new standard for open-source models and shows significant benefits from inference-time reasoning. In the future, we hope to incorporate more diverse domains of structured visuals that can be rendered via code, such as molecular formulas, musical staff, and even structured videos like educational videos.

## 7 ACKNOWLEDGMENTS

This study was supported in part by National Key R&D Program of China Project 2022ZD0161100, 2022ZD0115502, in part by the Centre for Perceptual and Interactive Intelligence, a CUHK-led InnoCentre under the InnoHK initiative of the Innovation and Technology Commission of the Hong Kong Special Administrative Region Government, in part by NSFC-RGC Project N_CUHK498/24, Ningbo Science and Technology Innovation 2025 Major Project (2025Z034), National Natural Science Foundation of China (NO. 62461160308, U23B2010), "Pioneer" and "Leading Goose" R&D Program of Zhejiang (No. 2024C01161). In addition, this research was supported in part by NUS Start-up Grant A-0010106-00-00 and in part by Guangdong Basic and Applied Basic Research Foundation (No. 2023B1515130008, XW).

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

# A APPENDIX

## A.1 RELATED WORK

### A.1.1 GENERATIVE MODELS FOR VISUAL CONTENT

**Text-to-Image Generation.** The text-to-image (T2I) field has advanced rapidly in recent years, with diffusion models driving much of the progress. Early U-Net-based diffusion models (Rombach et al., 2022; Podell et al., 2023) have been followed by diffusion transformers (Peebles & Xie, 2022; Chen et al., 2023; Esser et al., 2024; Zhuo et al., 2024; Xie et al., 2025a; Labs, 2024; Qin et al., 2025; Xie et al., 2025b), which substantially improve scalability and fidelity. More recently, aided by advances in large language models and visual tokenizers, autoregressive image generation has narrowed the gap with diffusion-based methods (Sun et al., 2024; Liu et al., 2024; Xin et al., 2025; Han et al., 2025a). State-of-the-art T2I systems now produce high-resolution, photorealistic, and aesthetically appealing images, largely through scaling data and model capacity.

**Image Editing.** Instruction-based image editing has evolved in tandem with T2I. Some early approaches (Couairon et al., 2022; Pan et al., 2023; Yang et al., 2023) achieved training-free edits by manipulating or guiding the denoising trajectory during diffusion sampling, but these methods often lacked stability and robustness. Subsequent work (Brooks et al., 2023; Sheynin et al., 2024; Huang et al., 2024; Wei et al., 2024; Chen et al., 2024; Pu et al., 2025) has increasingly framed editing as controllable generation, constructing large-scale editing datasets to fine-tune pre-trained T2I backbones. Recently, systems such as GPT-Image (OpenAI, 2025b), Nano Banana (Google, 2025), and Bagel (Deng et al., 2025) have begun to unify visual generation, editing, and even visual understanding within a single framework. These unified models inherit world knowledge from language models and, through data-driven multimodal training, exhibit strong instruction following alongside emerging reasoning capabilities.

### A.1.2 DATASET AND BENCHMARKS FOR GENERATION AND EDITING

**Datasets.** Most T2I datasets (Meyer et al., 2024; Sun et al., 2023; Ye et al., 2025a; Fang et al., 2025b) consist of natural images, either filtered from real photographs using aesthetic scores or synthesized with open-source models. This bias has encouraged models to overfit aesthetic metrics, often yielding images with a conspicuous "AI look." Image-editing datasets (Wei et al., 2024; Ye et al., 2025b) are largely inspired by InstructPix2Pix (Brooks et al., 2023), where synthetic pairs are produced via expert models and carefully designed pipelines; other efforts extract editing pairs from frames in real or synthetic videos (Chen et al., 2025b; Chang et al., 2025). Consequently, these editing datasets can be imprecise, as both expert-generated content and video-derived pairs introduce substantial noise. In contrast, our dataset provides strict code–image alignment and enforces precise state transitions through explicit code-level editing operations, ensuring faithful and verifiable supervision for both generation and editing.

## A.2 TRAINING DETAILS

The training of our model is organized into three progressive stages as shown in Fig. 9. The hyperparameters for the three-stage training process are provided in Tab. 4. Throughout all stages, we incorporate a diverse mixture of image generation and editing datasets, spanning both natural and structured sources, which helps the model learn generalizable representations and effectively adapt to diverse editing scenarios. Specifically, simple text-to-image datasets (jackyhate, 2024; Chen et al., 2025a) and various image editing datasets (Chen et al., 2025a; Wang et al., 2025c; Ye et al., 2025b; Kuprashevich et al., 2025) are used for stage 1 training. For stage 2 training, we further incorporate editing and text-to-image data from our structured dataset, along with a recent high-quality text-to-image dataset (Fang et al., 2025b). For stage 3 training, we use the same data sources as stage 2, but replace the short prompt with the detailed chain-of-thought instruction.

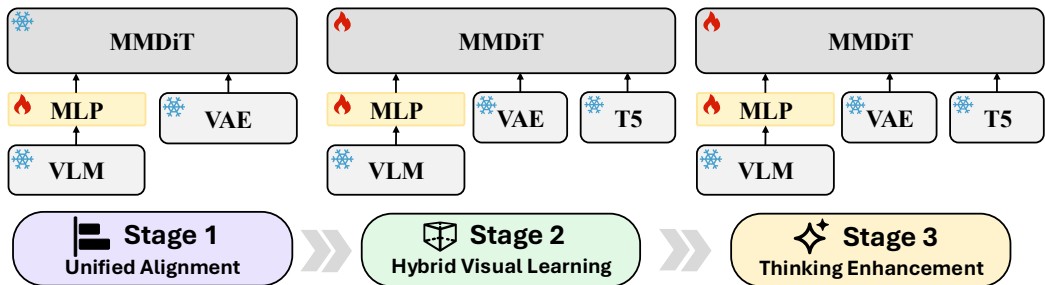

Figure 9: **Progressive training pipeline.** For the Unified Alignment stage, only newly added MLP parameters are updated. Both MMDiT and MLP parameters are updated during the Hybrid Visual Learning and Thinking Enhancement stages.

Table 4: **Training hyper-parameters for our model.**

| Hyper-parameters | Stage 1 | Stage 2 | Stage 3 |
|---|---|---|---|
| trainable parameters | MLP | MLP + DiT | MLP + DiT |
| warmup steps | 500 | 100 | 0 |
| learning rate schedule | | constant | |
| optimizer | | AdamW | |
| optimizer hyper-parameters | | $\beta_1, \beta_2 = (0.9, 0.95)$ | |
| weight decay | | 0.1 | |
| gradient norm clip | | 0.5 | |
| deepspeed stage | zero1 | zero2 | zero2 |
| learning rate | 5e-4 | 1e-5 | 1e-5 |
| batch size | 1024 | 512 | 512 |
| max text length | 256 | 256 | 512 |

## A.3 ADDITIONAL EVALUATION RESULTS

**Using Qwen2.5-VL-72B as evaluator.** With the rapid improvement of open-source models, it has become increasingly feasible to adopt them as cost-effective evaluators. Therefore, we provide the full benchmark results using open-source Qwen2.5-VL-72B (Bai et al., 2025) as our evaluator.

The ranking trends evaluated by GPT-5 (OpenAI, 2025a) and Qwen2.5-VL are generally well aligned. Both evaluators reliably highlight the strongest models (e.g., Nano Banana (Google, 2025), Seedream 4.0 (Seedream et al., 2025)) and the weakest one (e.g., UniWorld-V1 (Lin et al., 2025)). And we observe that slight discrepancies arise only among closely performing models. For example, GPT-5 slightly favors Seedream 4.0 over Nano Banana (shown in Tab .1), whereas Qwen2.5-VL shows the opposite trend in Tab.5. A similar alignment holds on the StructT2IBench, where both evaluations yield nearly identical overall rankings, with only minor variations among the top-performing models (Tab. 2 vs. 7). This alignment indicates that current open-source models have the potential to promote broader adoption of open-source evaluation within the community, reducing reliance on proprietary systems while maintaining reliable assessment quality.

**Different metrics analysis.** As shown in Tab. 5, Tab. 6 and Tab. 7, we observe that previously widely used metrics such as PSNR and SSIM are not reliable in structured visuals, since evaluation here emphasizes semantic-level correctness rather than purely pixel-level correspondence. Both GPT-5 and Qwen2.5-VL evaluators consistently reflect this phenomenon. To further validate this finding, we conduct a human alignment study, with detailed results presented in Fig. 11.

**General editing benchmark evaluation.** To assess the effectiveness of our proposed multi-stage training pipeline in preserving model capabilities in general domains, we provide the results evaluated on ImgEdit (Ye et al., 2025b) benchmark in Tab. 8. Our model achieves results competitive with state-of-the-art systems and outperforms the original FLUX.1 Kontext, validating that the training pipeline improves editing competence without sacrificing general-domain capability. We provide additional qualitative samples from ImgEdit benchmark in Fig. 10

Table 5: **Quantitative comparison on StructEditBench with Qwen2.5-VL-72B evaluator.** The table showcases the results of Accuracy (%), PSNR and SSIM (with GT image) for various methods. ■ ■ indicate the first and second Accuracy results, respectively.

| Model | Math | | | Chart | | | Graph | | | Puzzle | | | Science | | | Table | | | Overall | | |
|---|---|---|---|---|---|---|---|---|---|---|---|---|---|---|---|---|---|---|---|---|---|
| | Acc↑ | PSNR↑ | SSIM↑ | Acc↑ | PSNR↑ | SSIM↑ | Acc↑ | PSNR↑ | SSIM↑ | Acc↑ | PSNR↑ | SSIM↑ | Acc↑ | PSNR↑ | SSIM↑ | Acc↑ | PSNR↑ | SSIM↑ | Acc↑ | PSNR↑ | SSIM↑ |
| GPT-Image | 39.77 | 17.06 | 0.9199 | 39.61 | 12.56 | 0.658 | 47.77 | 17.24 | 0.9357 | 59.01 | 16.55 | 0.9139 | 53.23 | 16.61 | 0.9293 | 78.02 | 14.35 | 0.84 | 44.00 | 16.64 | 0.9119 |
| Nano Banana | 39.46 | 20.77 | 0.9526 | 42.33 | 14.93 | 0.7392 | 46.72 | 21.22 | 0.9616 | 54.35 | 22.92 | 0.9621 | 53.33 | 22.61 | 0.9646 | 66.87 | 19.75 | 0.926 | 44.46 | 21.09 | 0.9524 |
| Seedream 4.0 | 41.31 | 23.63 | 0.9587 | 43.17 | 16.54 | 0.7992 | 41.18 | 24.12 | 0.9433 | 59.94 | 26.93 | 0.9384 | 53.39 | 26.46 | 0.9518 | 78.56 | 24.75 | 0.9383 | 46.02 | 24.45 | 0.9514 |
| Uniworld | 15.97 | 8.84 | 0.8264 | 6.19 | 7.87 | 0.4883 | 10.27 | 6.16 | 0.7272 | 9.98 | 7.71 | 0.7619 | 16.01 | 7.67 | 0.7672 | 15.08 | 8.24 | 0.7297 | 9.98 | 8.21 | 0.7895 |
| DiMOO | 22.22 | 21.56 | 0.9519 | 8.78 | 14.98 | 0.7437 | 16.30 | 21.77 | 0.96 | 21.18 | 22.26 | 0.9568 | 18.11 | 22.57 | 0.9626 | 20.60 | 19.47 | 0.9209 | 14.22 | 21.49 | 0.9504 |
| OmniGen2 | 23.65 | 15.95 | 0.8206 | 13.44 | 12.44 | 0.7187 | 23.68 | 11.31 | 0.7834 | 16.56 | 16.51 | 0.8654 | 26.02 | 15.60 | 0.8615 | 27.52 | 16.60 | 0.8888 | 18.10 | 15.49 | 0.8334 |
| Ovis-U1 | 25.57 | 18.45 | 0.9356 | 13.48 | 13.30 | 0.6854 | 28.42 | 19.01 | 0.9501 | 24.59 | 17.92 | 0.924 | 22.00 | 18.68 | 0.9455 | 33.03 | 16.62 | 0.8796 | 19.40 | 18.25 | 0.93 |
| Hidream-E1.1 | 21.97 | 18.43 | 0.9399 | 20.25 | 12.91 | 0.7059 | 22.40 | 18.26 | 0.9466 | 29.16 | 18.04 | 0.9389 | 25.42 | 16.47 | 0.9377 | 46.83 | 17.12 | 0.9118 | 23.07 | 18.01 | 0.9368 |
| Bagel | 16.07 | 21.38 | 0.9603 | 24.27 | 16.38 | 0.8178 | 20.32 | 22.70 | 0.9727 | 35.44 | 24.22 | 0.9728 | 32.58 | 23.56 | 0.9736 | 45.37 | 21.54 | 0.9542 | 24.24 | 22.06 | 0.9636 |
| Bagel-Think | 27.82 | 23.97 | 0.9734 | 23.38 | 16.82 | 0.8227 | 32.30 | 26.49 | 0.9823 | 23.22 | 26.75 | 0.9809 | 33.19 | 25.57 | 0.9804 | 35.73 | 23.83 | 0.9731 | 26.17 | 24.70 | 0.9759 |
| Step1X-Edit | 27.22 | 23.41 | 0.9711 | 24.99 | 16.68 | 0.8154 | 25.11 | 24.56 | 0.9775 | 43.95 | 25.94 | 0.9786 | 33.49 | 24.98 | 0.9738 | 48.48 | 23.97 | 0.9687 | 28.26 | 24.03 | 0.9726 |
| Flux Kontext | 29.78 | 19.78 | 0.9512 | 27.78 | 14.61 | 0.7324 | 30.25 | 20.10 | 0.9554 | 46.32 | 20.38 | 0.9501 | 35.60 | 20.99 | 0.9627 | 52.22 | 18.52 | 0.9127 | 31.13 | 19.84 | 0.9478 |
| Qwen-Edit | 34.03 | 23.73 | 0.9714 | 25.50 | 12.33 | 0.6593 | 38.23 | 26.11 | 0.9814 | 43.25 | 27.31 | 0.9794 | 43.08 | 25.53 | 0.9742 | 58.82 | 25.71 | 0.9665 | 31.99 | 24.81 | 0.9731 |
| Ours | 41.10 | 23.31 | 0.9514 | 40.44 | 15.33 | 0.7528 | 37.20 | 24.65 | 0.9626 | 57.58 | 26.33 | 0.9647 | 52.46 | 25.80 | 0.9638 | 69.75 | 23.19 | 0.9438 | 43.56 | 24.01 | 0.8535 |

Table 6: **Quantitative comparison of different methods on StructEditBench (Chart only) with Qwen2.5-VL-72B evaluator.** ■ ■ indicate the first and second Accuracy results, respectively.

| Model | Category | | | Color | | | Num | | | Auxiliary | | | Add&Del | | | Overall | | |
|---|---|---|---|---|---|---|---|---|---|---|---|---|---|---|---|---|---|---|
| | Acc↑ | PSNR↑ | SSIM↑ | Acc↑ | PSNR↑ | SSIM↑ | Acc↑ | PSNR↑ | SSIM↑ | Acc↑ | PSNR↑ | SSIM↑ | Acc↑ | PSNR↑ | SSIM↑ | Acc↑ | PSNR↑ | SSIM↑ |
| GPT-Image | 37.65 | 9.85 | 0.6756 | 45.26 | 13.91 | 0.6802 | 31.33 | 13.48 | 0.6598 | 50.63 | 13.48 | 0.6315 | 33.56 | 12.73 | 0.6386 | 39.61 | 12.56 | 0.6580 |
| Nano Banana | 37.13 | 10.33 | 0.6728 | 47.56 | 17.59 | 0.8117 | 38.04 | 16.56 | 0.7696 | 55.05 | 17.39 | 0.7593 | 35.72 | 13.88 | 0.6989 | 42.33 | 14.93 | 0.7392 |
| Seedream 4.0 | 34.76 | 9.67 | 0.6668 | 54.75 | 21.77 | 0.9162 | 39.62 | 19.22 | 0.8509 | 55.69 | 19.46 | 0.8490 | 35.16 | 14.28 | 0.7457 | 43.17 | 16.54 | 0.7992 |
| Uniworld | 6.52 | 7.60 | 0.5067 | 7.83 | 8.29 | 0.5542 | 4.71 | 7.57 | 0.4504 | 8.94 | 8.62 | 0.4758 | 2.93 | 7.35 | 0.4496 | 6.19 | 7.87 | 0.4883 |
| DiMOO | 5.75 | 9.82 | 0.6559 | 8.34 | 16.97 | 0.8120 | 11.40 | 17.30 | 0.7790 | 9.99 | 17.59 | 0.7770 | 9.61 | 14.46 | 0.7160 | 8.78 | 14.98 | 0.7437 |
| OmniGen2 | 12.41 | 8.61 | 0.6579 | 21.35 | 11.78 | 0.7786 | 13.91 | 14.03 | 0.7138 | 13.08 | 16.59 | 0.7839 | 8.35 | 12.11 | 0.6735 | 13.44 | 12.44 | 0.7187 |
| Ovis-U1 | 11.11 | 9.57 | 0.6567 | 20.34 | 15.38 | 0.7456 | 13.33 | 15.13 | 0.7167 | 13.27 | 14.25 | 0.6541 | 11.38 | 13.11 | 0.6612 | 13.48 | 13.30 | 0.6854 |
| Hidream-E1.1 | 18.14 | 9.29 | 0.6438 | 30.63 | 14.07 | 0.7667 | 18.60 | 15.04 | 0.7418 | 22.39 | 14.34 | 0.7135 | 13.86 | 12.71 | 0.6788 | 20.25 | 12.91 | 0.7059 |
| Bagel-Think | 20.19 | 9.08 | 0.6592 | 36.68 | 20.46 | 0.9482 | 24.15 | 20.29 | 0.8845 | 23.06 | 21.24 | 0.8888 | 16.42 | 14.93 | 0.7730 | 23.38 | 16.82 | 0.8227 |
| Bagel | 18.31 | 9.00 | 0.6714 | 40.57 | 19.35 | 0.9356 | 24.79 | 20.14 | 0.8760 | 26.19 | 20.62 | 0.8764 | 16.45 | 14.58 | 0.7654 | 24.27 | 16.38 | 0.8178 |
| Step1X-Edit | 19.80 | 10.40 | 0.6780 | 33.99 | 19.75 | 0.9202 | 26.52 | 20.22 | 0.8719 | 26.38 | 19.46 | 0.8692 | 21.77 | 15.11 | 0.7712 | 24.99 | 16.68 | 0.8154 |
| Qwen-Edit | 19.16 | 9.52 | 0.6841 | 34.22 | 13.63 | 0.6850 | 22.89 | 13.69 | 0.6726 | 32.19 | 13.07 | 0.6048 | 22.17 | 12.46 | 0.6440 | 25.50 | 12.33 | 0.6593 |
| Flux Kontext | 21.70 | 10.14 | 0.6832 | 36.89 | 16.53 | 0.7896 | 27.77 | 16.74 | 0.7547 | 33.81 | 16.79 | 0.7442 | 22.26 | 13.95 | 0.7022 | 27.78 | 14.61 | 0.7324 |
| Ours | 39.55 | 10.40 | 0.7075 | 50.46 | 18.16 | 0.8257 | 32.39 | 17.04 | 0.7721 | 48.99 | 17.90 | 0.7685 | 31.73 | 14.37 | 0.7012 | 40.44 | 15.33 | 0.7528 |

## A.4 DETAILED HUMAN ALIGNMENT STUDY

**Details of Elo rating.** We leverage Rapidata[2] to collect human preference annotations. We compensated all annotators at or above the local minimum wage. After collecting human preferences, we adopt a robust Elo rating scheme to analyze the results of human studies. For a match between model $A$ and model $B$ with current ratings $(R_A, R_B)$, the expected win probability of $A$ is defined as:

$$E_A = \frac{1}{1 + 10^{\frac{R_B - R_A}{\sigma}}}, \tag{1}$$

where $\sigma = 400$ is the scaling factor.

Given the empirical vote proportion $s_A \in [0, 1]$ for model $A$ (with $s_B = 1 - s_A$), the ratings are updated as:

$$R'_A = \max(R_{\min}, R_A + K_{\text{eff}}(s_A - E_A)),$$
$$R'_B = \max(R_{\min}, R_B + K_{\text{eff}}(s_B - E_B)), \tag{2}$$

where $K_{\text{eff}} = K \cdot \frac{V}{5}$ is the effective step size, scaled by the number of votes $V$, $R_{\min} = 700$ is the rating floor, and $K = 24$ is the base learning rate.

To improve robustness, we shuffle the match order and repeat the Elo computation for $T = 50$ rounds. The final Elo score for each model $m$ is reported as the mean and standard deviation across rounds:

$$\bar{R}_m = \frac{1}{T}\sum_{t=1}^{T} R_m^{(t)}, \qquad \sigma_m = \sqrt{\frac{1}{T}\sum_{t=1}^{T}\left(R_m^{(t)} - \bar{R}_m\right)^2}. \tag{3}$$

This procedure accounts for vote strength, enforces a minimal rating floor, and provides both robust mean scores and uncertainty estimates. The parameter setting is shown in Tab. 9

---

[2]https://www.rapidata.ai/

Table 7: **Quantitative comparison on StructT2IBench with Qwen2.5-VL-72B evaluators**, reporting Accuracy (%).

| Model | Chart | Graph | Math | Puzzle | Science | Table | Overall |
|---|---|---|---|---|---|---|---|
| Seedream 4.0 | 32.07 | 48.38 | **55.62** | 42.81 | **54.43** | 65.61 | 42.33 |
| Nano banana | 33.26 | **52.00** | 54.87 | **53.26** | 47.81 | 65.20 | 43.16 |
| GPT-Image | **34.00** | 51.47 | 53.12 | 51.05 | 52.52 | **79.26** | **44.08** |
| Bagel | 2.33 | 0.78 | 1.46 | 2.51 | 5.19 | 2.94 | 2.21 |
| Bagel-Think | 2.83 | 10.56 | 8.87 | 11.94 | 11.76 | 8.35 | 5.93 |
| Uniworld | 4.66 | 11.82 | 10.45 | 8.01 | 14.14 | 9.85 | 7.40 |
| Hidream-I1-Full | 5.07 | 16.57 | 14.29 | 13.61 | 22.65 | 24.69 | 10.39 |
| OmniGen2 | 8.62 | 19.23 | 16.93 | 14.51 | 22.63 | 19.96 | 12.87 |
| Flux1.1 Dev | 10.28 | 18.84 | 17.43 | 18.64 | 19.06 | 24.66 | 14.19 |
| Fulx Kontext | 14.81 | 20.60 | 16.73 | 21.68 | 24.54 | 26.44 | 17.09 |
| Ovis-U1 | 29.33 | 11.79 | 11.74 | 17.89 | 18.02 | 10.21 | 21.81 |
| Qwen-Image | 29.58 | 42.99 | 41.74 | 42.32 | 45.14 | 69.00 | 37.03 |
| Ours | 15.90 | 23.83 | 31.87 | 24.87 | 31.69 | 27.81 | 22.12 |

Table 8: **Quantitative comparison on ImgEdit-Full benchmark.**

| Model | Add | Adjust | Extract | Replace | Remove | Background | Style | Hybrid | Action | Overall |
|---|---|---|---|---|---|---|---|---|---|---|
| MagicBrush | 2.84 | 1.58 | 1.51 | 1.97 | 1.58 | 1.75 | 2.38 | 1.62 | 1.22 | 1.90 |
| Instruct-Pix2Pix | 2.45 | 1.83 | 1.44 | 2.01 | 1.50 | 1.44 | 3.55 | 1.20 | 1.46 | 1.88 |
| AnyEdit | 3.18 | 2.95 | 1.88 | 2.47 | 2.23 | 2.24 | 2.85 | 1.56 | 2.65 | 2.45 |
| UltraEdit | 3.44 | 2.81 | 2.13 | 2.96 | 1.45 | 2.83 | 3.76 | 1.91 | 2.98 | 2.70 |
| OmniGen | 3.47 | 3.04 | 1.71 | 2.94 | 2.43 | 3.21 | 4.19 | 2.24 | 3.38 | 2.96 |
| Step1X-Edit | 3.88 | 3.14 | 1.76 | 3.40 | 2.41 | 3.16 | 4.63 | 2.64 | 2.52 | 3.06 |
| ICEdit | 3.58 | 3.39 | 1.73 | 3.15 | 2.93 | 3.08 | 3.84 | 2.04 | 3.68 | 3.05 |
| BAGEL | 3.56 | 3.31 | 1.70 | 3.30 | 2.62 | 3.24 | 4.49 | 2.38 | 4.17 | 3.20 |
| UniWorld-V1 | 3.82 | 3.64 | 2.27 | 3.47 | 3.24 | 2.99 | 4.21 | 2.96 | 2.74 | 3.26 |
| OmniGen2 | 3.57 | 3.06 | 1.77 | 3.74 | 3.20 | 3.57 | 4.81 | 2.52 | 4.68 | 3.44 |
| FLUX.1 Kontext | 3.76 | 3.45 | 2.15 | 3.98 | 2.94 | 3.78 | 4.38 | 2.96 | 4.26 | 3.52 |
| GPT-Image | 4.61 | 4.33 | 2.90 | 4.35 | 3.66 | 4.57 | 4.93 | 3.96 | 4.89 | 4.20 |
| Qwen-Image | 4.38 | 4.16 | 3.43 | 4.66 | 4.14 | 4.38 | 4.93 | 3.82 | 4.69 | 4.27 |
| **Ours** | 4.15 | 4.07 | 1.62 | 4.01 | 3.60 | 4.01 | 4.52 | 3.58 | 4.22 | 3.75 |

Table 9: **Elo parameter setting.**

| Parameter | Number |
|---|---|
| initial Elo mean | 1,000 |
| Elo standard deviation | 300 |
| base of logarithm | 10 |
| scaling factor | 400 |
| K-factor | 24 |
| minimum Elo rating | 700 |
| number of simulated matches | 4,500 |

**Results analysis.** As depicted in Fig. 11, StructScore Accuracy demonstrates a much stronger correlation with human evaluation results than conventional metrics such as PSNR and SSIM. Specifically, StructScore achieves a Pearson correlation of $r = 0.87$ at the overall level and $r = 0.84$ on chart-specific cases, substantially higher than PSNR ($r = 0.69/0.64$) and SSIM ($r = 0.70/0.55$). These results clearly indicate that StructScore better captures the semantic-level fidelity that aligns with human preference, whereas pixel-level metrics fail to reflect such perceptual correctness. However, compared with GPT-5 evaluators' $r = 0.92$ (shown in Fig. 6), the alignment of Qwen2.5-VL with human evaluation is still weaker. This is probably because its performance on such structured images remains inferior to closed-source models, which could be improved as open-source models continue to advance.

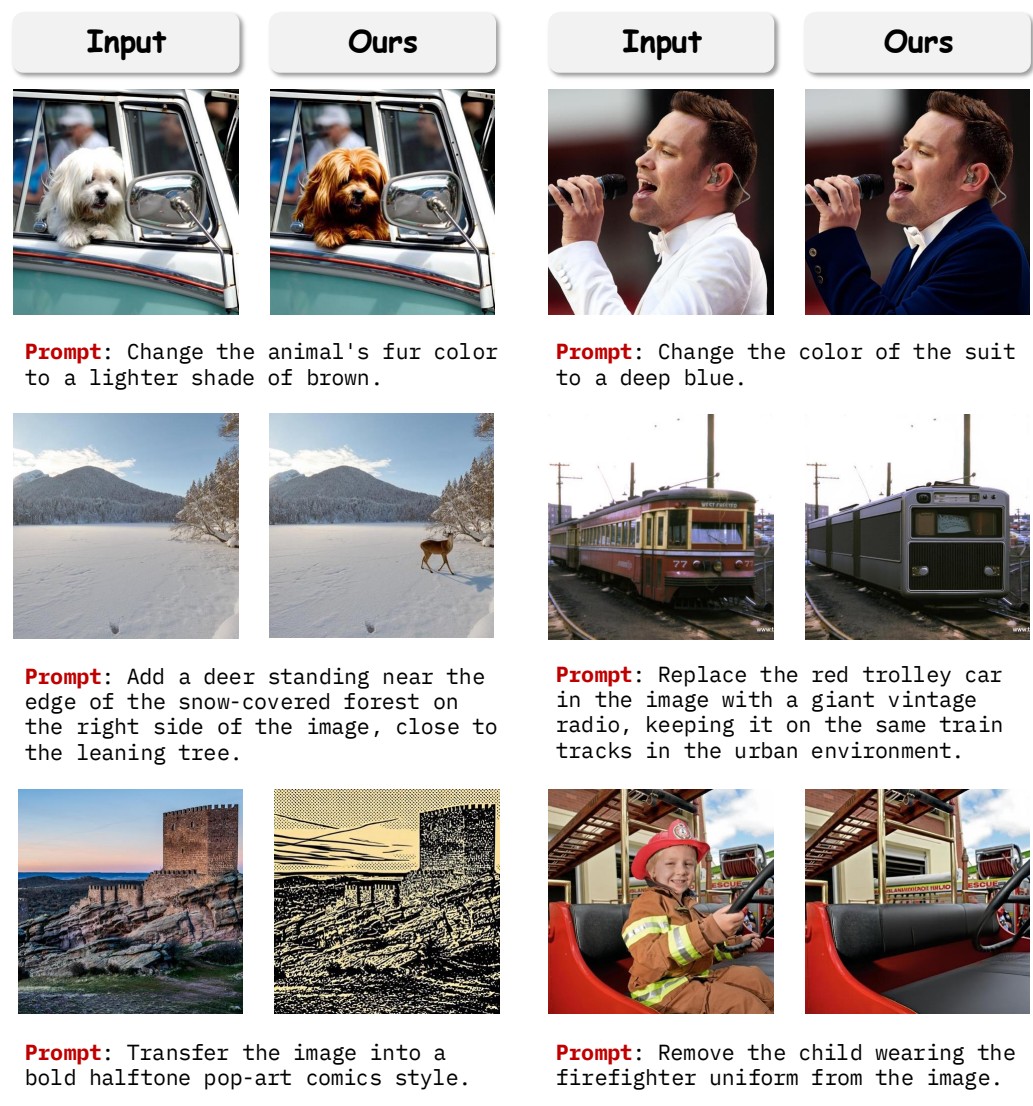

Figure 10: **Qualitative results of our model on ImgEdit-Full benchmark.**

## A.5 Ablation on different weight setting

We conducted a comparative study of benchmark ranking calculated with different weight ratios by evaluating their alignment with human preference judgments. As illustrated in Fig. 12, the configuration with a 1:9 weighting achieved the highest correction coefficient, followed by the 2:8 setting. In contrast, the 3:7 and 4:6 ratios yielded substantially lower values. These findings demonstrate that the 1:9 ratio most accurately captures the alignment between the model's evaluation accuracy and human preference consistency.

## A.6 Case Study

For qualitative evaluation, we provide text-to-image generation and image editing examples across various models in Fig. 13 and 14. Utilizing Nano Banana 2, we generate a collection of structurally complex and more realistic images, on which our model only trained on code-rendered images exhibits generalization capability (shown in Fig. 16). These results demonstrate that the proposed dataset effectively promotes the advancement of unified models within the structured image domain.

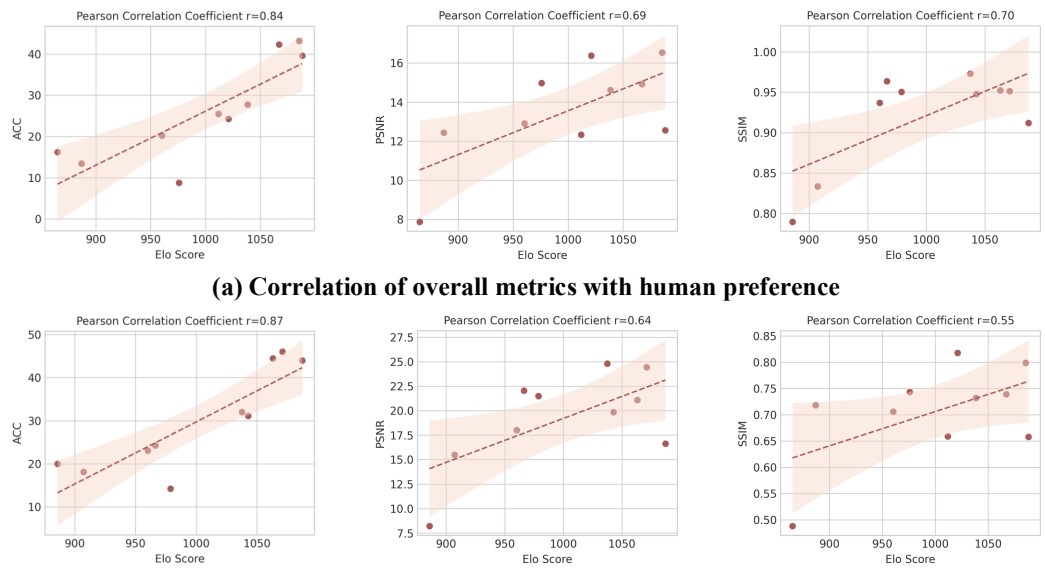

Figure 11: **Pearson correlation analysis** among Accuracy (Qwen2.5-VL-72B evaluator), PSNR, SSIM and human Elo score for image editing.

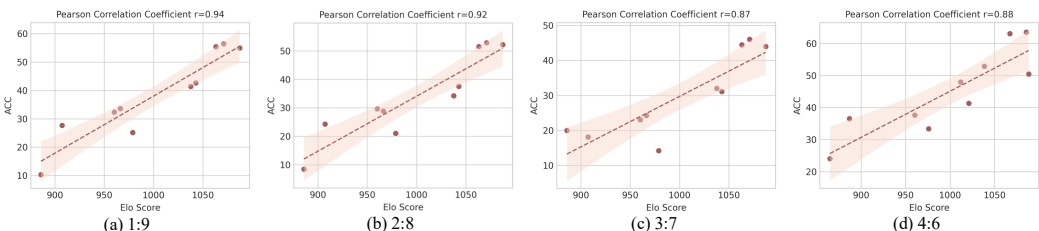

Figure 12: **Correlation of our metric with human preference under different weighting ratios.**

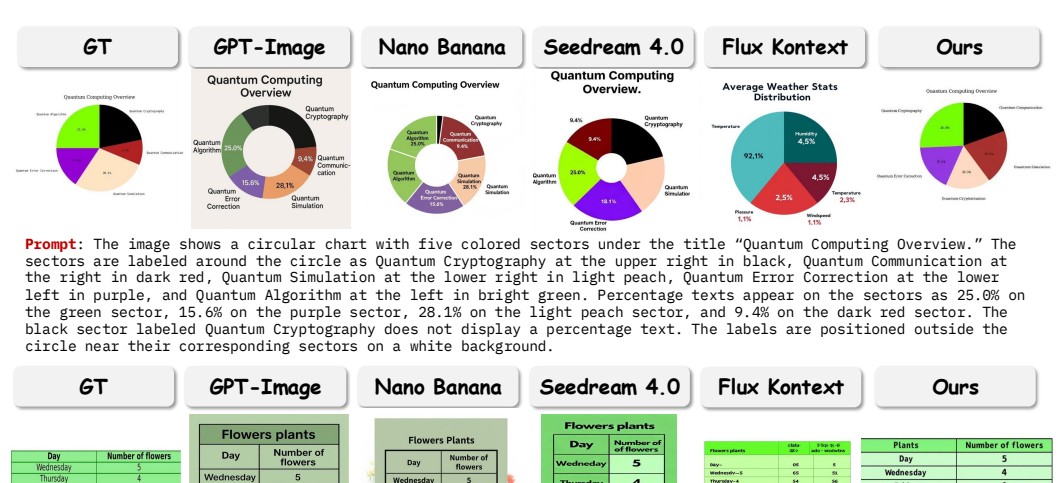

Figure 14: **Qualitative comparison for structured image generation.** "GT" stands for ground truth.

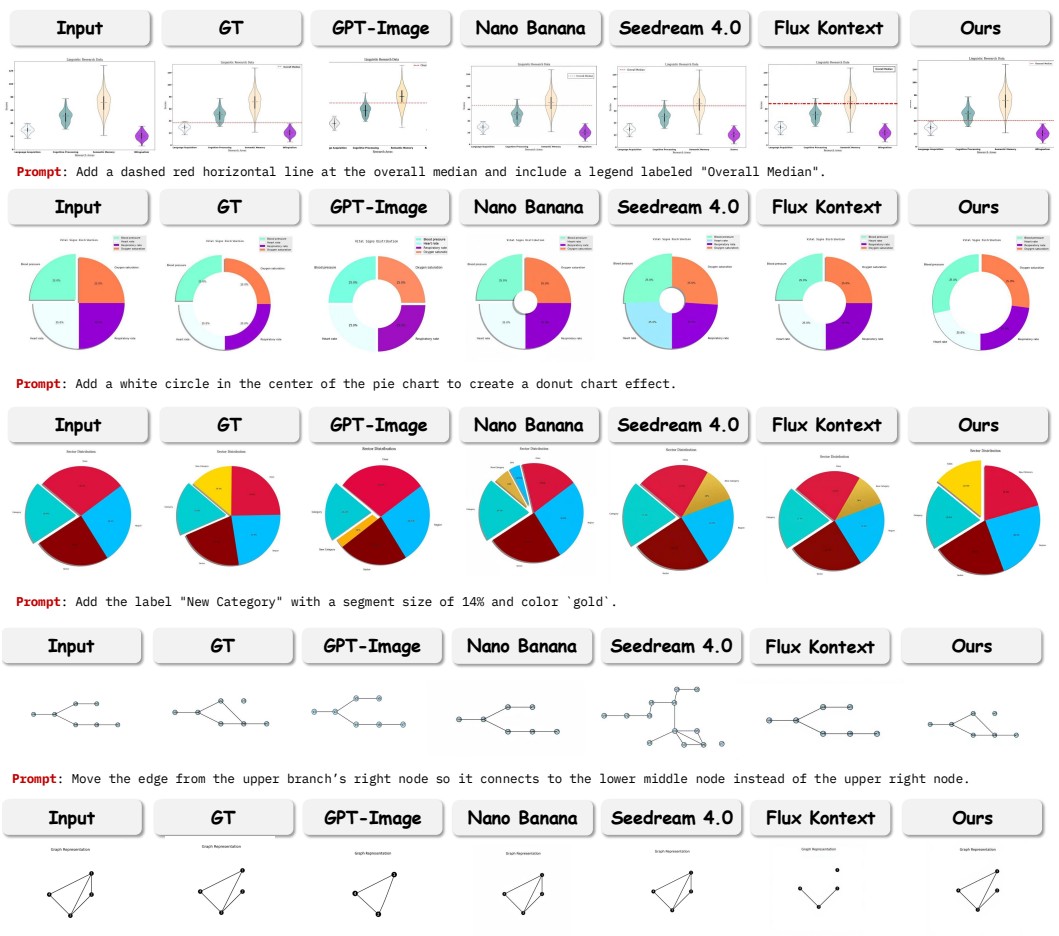

**Prompt**: Add a dashed red horizontal line at the overall median and include a legend labeled "Overall Median".

**Prompt**: Add a white circle in the center of the pie chart to create a donut chart effect.

**Prompt**: Add the label "New Category" with a segment size of 14% and color `gold`.

**Prompt**: Move the edge from the upper branch's right node so it connects to the lower middle node instead of the upper right node.

**Prompt**: Edit instruction: Remove the single vertical line segment connecting node 1 to node 2.

Figure 13: **Qualitative comparison for structured image editing.** "GT" stands for ground truth.

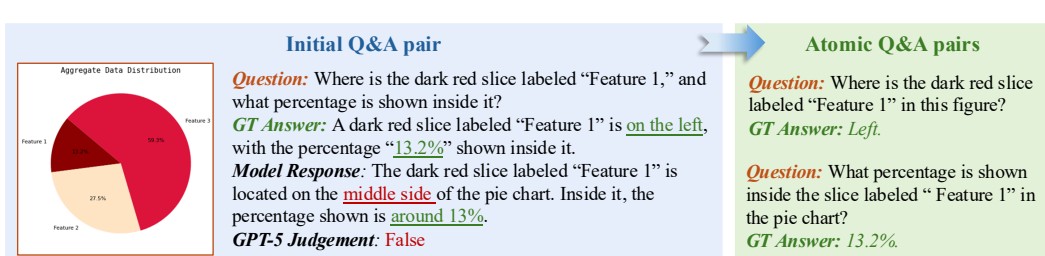

Figure 15: **Comparison of the initial and revised atomic Q&A pairs**. Initial Q&A pairs sometimes entangle multiple attributes, hindering unambiguous verification and accurate scoring. Enforcing atomicity, *i.e.*, one attribute or relation per Q&A, substantially improves metric reliability.

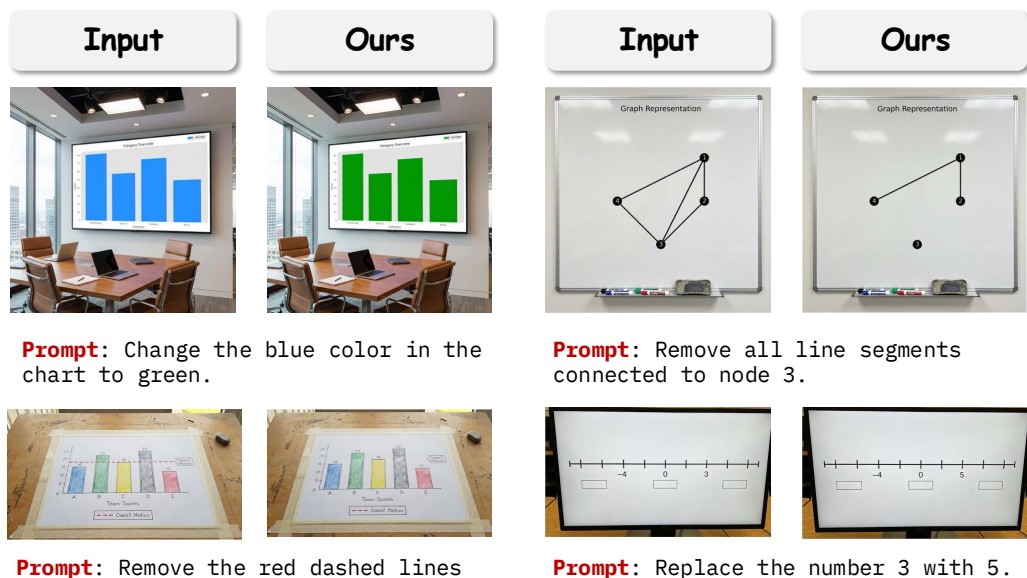

**Prompt**: Change the blue color in the chart to green.

**Prompt**: Remove all line segments connected to node 3.

**Prompt**: Remove the red dashed lines from the chart.

**Prompt**: Replace the number 3 with 5.

Figure 16: **Qualitative results of the generalization capability of our model.**

A.7    PROMPTS

In this section, we present the complete set of prompts utilized in the data generation pipeline, alongside those employed for evaluation. Specifically, these include the prompt designed for editing pair construction in Fig. 17, the prompt for generating the T2I prompts in Fig. 18, the prompt used for constructing CoT in Fig. 19, the prompt applied for benchmark description construction in Fig. 20, the prompt for benchmark Q&A construction in Fig. 21, the prompt for benchmark Q&A refinement in Fig. 22,the prompt for GT judgement in Fig. 23.

---

**Editing Pair Construction**

You will be given: (a) a STEM-related figure, and (b) the source code that renders it.
Your goal is to analyze the figure and produce one salient feature and a matched pair of edit commands by strictly following these steps:

**1.  Identify Salient Feature:**
   * First, internally determine the image's academic discipline or content domain (e.g., geometry, mathematical chart, puzzle game, physics diagram, photograph, etc.).
   * Then, based on the identified domain, describe its most core and relevant characteristic in one concise sentence.
      * For a geometric figure, describe a key "geometric feature."
      * For a mathematical chart, describe its core "mathematical property" (such as a trend or relationship).
      * For a puzzle or game, describe a key "game state or rule."
      * For other specialized images, identify the domain-related core content.

**2.  Produce Edit Commands:**
   * Based on the feature above, provide a pair of concise, direct instructions (code edit command, image edit command) that modify one core semantic element to create a new (code, image) pair.
   * Use diverse editing operations (e.g., add, remove, replace, move, rotate, scale, etc.) to modify diverse attributes of the diverse elements in the image.
   * For code edit command, provide a precise and executable instruction to modify the source code to achieve the desired change.
   * For image edit command, derive it only from the visual content in the image. Do not mention specific content in the source code, such as exact numeric values.
   * The two commands must strictly correspond: executing the code edit should produce the stated image change on exactly one element.

Your output **must strictly adhere** to the format below, containing no additional text or explanations.

```
Salient Feature: [A single sentence describing the most relevant, domain-specific core feature]
Code Edit Command: [one precise, code-level instruction that implements the image edit]
Image Edit Command: [one concise, visual-only instruction for a single element]
```

Figure 17: **Prompt constructing the image editing pairs with code commands.**

---

**T2I Prompt Construction**

You are an expert describer of non-natural STEM visuals (e.g., plots, charts, diagrams, schematics, equations, geometric figures, etc.). Given an image, produce a single, self-contained caption that enables a reader to understand the content without seeing the image.

**Requirements:**
1. Identify the type of visual and its main concept or purpose.
2. Describe all visible elements with their structure and relationships.
3. For each element, precisely describe its attributes like positions, values, symbols, text labels, colors, scales, and other annotations, if applicable.
4. Be specific and technical where appropriate, using proper STEM terminology.
5. Use concise, well-structured sentences or short paragraphs, not bullet lists.
6. Do not mention non-existing elements in the image.

Provide **ONLY** the caption text as a coherent string within **200 words**, no additional commentary.

---

Figure 18: **Prompt to obtain captions for non-natural STEM data.**

---

**CoT construction**

You are an expert image editor analyzing an editing task. Given a source image, a target image, and an editing instruction, annotate Chain-of-Thought reasoning step by step for the edit.

Step 1: Describe the key elements and their notable attributes in the source image.
Step 2: Identify the elements or areas in the source image that need to be edited and precisely predict the specific changes to make.
Step 3: Describe the final edited image, especially focusing on the modified content and how it integrates with the unchanged parts of the source image to achieve the target result.

Make sure the content in each step is concise but information-dense. Besides, make sure the information in each step is distinct and no overlapping.

**Editing Instruction:** {edit_command}

Output **ONLY** in this exact JSON format, with no additional text before or after. Ensure your response is a valid JSON object:

```
{{
"step1": "Your description for Step 1",
"step2": "Your identification for Step 2",
"step3": "Your description for Step 3",
}}
```

---

Figure 19: **Prompt to obtain reasoning traces for the editing task of non-natural STEM images.**

---

**Benchmark description construction**

You are an expert describer of non-natural STEM visuals (e.g., plots, charts, diagrams, schematics, equations, geometric figures, etc.). Given an image, produce a single, self-contained caption that enables a reader to understand the content without seeing the image.

Caption the provided target image by following these requirements:

Edit instruction: {edit_instruction}.

1. Identify the visual type and its primary purpose.

2. Describe all visible elements, their structure, and relationships.

3. Include explicit attributes such as positions, values, text labels, colors, scales, and annotations.

4. Use concise, well-structured sentences with technical terminology when appropriate.

5. Do not invent elements that are not visible.

Return only the caption text as a single coherent paragraph within 200 words.

Figure 20: **Prompt for evaluation.**

---

**Benchmark Q&A construction**

Convert the following image description into Q&A format by breaking down EVERY piece of information into the most detailed Q&A pairs possible.

**Instructions:**

1. Break down EACH sentence into its smallest meaningful information units

2. For compound sentences or lists, create SEPARATE Q&A pairs for EACH individual item or fact

3. Keep answers as concise as possible - use only the essential words/numbers needed

4. Format: "Q: [question]" followed by "A: [answer]" on the next line

5. Examples of how to break down:

- "The title of the plot is 'Distribution of Customs'" →

Q: What is the title of this plot?

A: Distribution of Customs

- "The numbers of A, B, C, D are 50, 70, 80, 100, respectively." →

Q: What is the number of A?

A: 50...

6. For lists or multiple items in one sentence, create individual Q&A pairs for EACH item

7. Remove unnecessary words from answers (like "The", "is", etc.) when possible while keeping meaning clear

8. Each Q&A pair should be immediately followed by the next with only a single line break between them

9. Do not skip any information from the original description

10. Do not add information not present in the original description

Image description: {caption}

Convert ALL information above into detailed Q&A pairs:

Figure 21: **Prompt for constructing question-answer pairs for our benchmark.**

---

**Benchmark Q&A Refinement**

Task: Rewrite the question and answer to be clear and unambiguous
The model gave an incorrect answer, likely because the original question was ambiguous or the answer didn't match what's actually in the image. Your task is to rewrite BOTH the question and answer while maintaining their original intent.
CRITICAL REQUIREMENTS:
Preserve the original intent - The rewritten Q&A must ask and answer about the SAME aspect/element as the original
Make the question unambiguous - Remove any vagueness so there's only ONE possible correct answer
Be specific about visual elements - Reference specific parts of the image (e.g., "the blue bar", "the leftmost column", "the value at point X")
Ensure the answer matches the question - The answer must directly correspond to what the question asks
Keep the answer concise and precise - Use minimal words while being accurate
Both must be verifiable - Anyone looking at the image should arrive at the same answer
Original Question: {question}
Original Answer: {ground_truth_answer}
Model's Incorrect Response: {model_response}
Rewriting Guidelines:
If asking about a value: Specify WHICH value (e.g., "What is the value of X at year 2020?" not just "What is the value?")
If asking about colors: Specify WHICH element's color (e.g., "What color is the bar for category A?")
If asking about comparisons: Specify WHAT is being compared (e.g., "Which month has the highest sales: Jan, Feb, or Mar?")
If asking about locations: Be precise (e.g., "In which quadrant is the red point located?")
Avoid pronouns without clear antecedents (it, this, that)
Avoid relative terms without reference points (high, low, large, small)
Examples of good rewrites:
Vague: "What is the highest value?" → Clear: "What is the highest value on the Y-axis?"
Vague: "What color is it?" → Clear: "What color is the line representing sales data?"
Vague: "Where is the peak?" → Clear: "At which X-coordinate does the blue line reach its maximum value?"
Output Format:
Output your response in JSON format with exactly these two keys:
{"rewritten_question": "[clear, unambiguous question about the SAME aspect as original]", "rewritten_answer": "[precise answer that matches the rewritten question and the image]"}
Remember: Keep the same intent as the original Q&A, just make it clearer and more precise!

Figure 22: **Prompt to refine the question-answer pairs.**

---

**VLM GT Judgement**

**Task: Evaluate if the model's response is correct.**
Based on the "Ground Truth Answer" and "Model Response" provided below, please determine if the "Model Response" is acceptable.
**Evaluation Criteria:**
Textual Part: The model's response should contain the key information from the "Ground Truth Answer".
Accept responses that include the correct answer, even with additional context or minor phrasing differences
Reject responses that provide fundamentally different or incorrect information
Case-insensitive comparison
Numerical Part: A tolerance of $\pm 10\%$ is allowed for any numerical values compared to the "Ground Truth Answer". You are expected to perform this calculation to check if the value is within the acceptable range.
**Inputs:**
**Output Requirements:**
Please judge strictly according to the rules above and output only the single word "Correct" or "Incorrect". Do not include any other explanations, reasons, or punctuation.

Figure 23: **Prompt to check if ground-truth and mdoel response are similar.**

