# OpenReview forum: "Factuality Matters: When Image Generation and Editing Meet Structured Visuals"
_ICLR.cc/2026/Conference — ICLR 2026 Poster_

### Official Review · Reviewer_vk2E · 2025-10-31

**Soundness:** 2
**Presentation:** 3
**Contribution:** 3
**Rating:** 6
**Confidence:** 3

**Summary:**

The paper first introduces a dataset for structured image editing, then propose a benchmark built from this dataset. It also trained a model using the dataset and evaluate on the benchmark.

**Strengths:**

1. Used programs to generate image, which makes it more robust and controlled.
2. The attempt to show alignment between human scores and benchmark metrics helps validate whether automated evaluation aligns with human judgment.

Overall, the data generation process seems reasonable for producing a large-scale benchmark, as the authors demonstrate that models trained from this pipeline show good performance both on this benchmark and on another relevant benchmarks.

**Weaknesses:**

1. The training pipeline can benefit from more ablation analysis. For example, what is the improvement when training with and without Stage 3 ? The choice of three stages is intuitive, but it requires more empirical evidence to justify the curriculum design. In addition, since the authors incorporate another dataset outside of their own dataset in stage 3, an ablation should be conducted to clearly show the effect of this addition, i.e. whether the improvement comes from the ChatGPT generated CoT or from the dataset itself.

2. The variation in evaluation results remains quite large with a different LLM(when comparing tables in the appendix produced by Qwen versus those in the main paragraph), even though the trends are similar. This is my main concern for the paper. Especially for a benchmark, if the score or model ranking can change depending on which VLM/LLM backend is used for scoring, it becomes difficult to trust this measure of progress. This subjectivity potentially suggests that the evaluation protocol may need further stabilization.

**Questions:**

Please see weakness

---

> ### Author Response · Authors · 2025-11-21
>
> We deeply appreciate your insightful comments and efforts in reviewing our manuscript. We respond to each of your comments one-by-one in what follows. In the revised draft, we mark our major revisions as “red”.
>
> > Q1: The training pipeline can benefit from more ablation analysis. For example, what is the improvement when training with and without Stage 3 ? The choice of three stages is intuitive, but it requires more empirical evidence to justify the curriculum design. In addition, since the authors incorporate another dataset outside of their own dataset in stage 3, an ablation should be conducted to clearly show the effect of this addition, i.e. whether the improvement comes from the ChatGPT generated CoT or from the dataset itself.
>
> Thank you for the insightful suggestion. We have added an ablation study comparing training **with and without Stage 3**. As shown in the following table, removing Stage 3 leads to a clear performance drop across all categories (while still remaining above the FLUX-Kontext baseline). This demonstrates the effectiveness of the progressive curriculum and highlights that **Stage 3 plays a crucial role by enabling the model to benefit from explicit reasoning at inference time.**
>
> Regarding the dataset used in Stage 3, we would like to clarify that Stages 2 and 3 use the same dataset. The additional data beyond our dataset are included solely to prevent degradation on natural images (as validated in Table 8 and Fig. 10). Therefore, the gains on our StructBench in Stage 3 come from the **GPT-5–generated reasoning chains**, which are consistent with our analysis in Sec. 5.3 that explicit reasoning significantly enhances performance on complex structured editing tasks.
>
> | Model           | Math  | Chart | Graph | Puzzle | Science | Table | Overall |
> |-----------------|------:|------:|------:|------:|-------:|------:|-------:|
> | FLUX.1 Kontext  | 31.33 | 32.39 | 40.21 | 57.47 | 49.39  | 57.04 | 37.8   |
> | Ours–Stage2     | 35.73 | 36.51 | 43.06 | 60.27 | 50.42  | 59.2  | 41.72  |
> | Ours            | 52.47 | 48.86 | 58.10  | 71.37 | 74.05  | 75.08 | 53.61  |

---

> > ### Author Response · Authors · 2025-11-21
> >
> > > Q2: The variation in evaluation results remains quite large with a different LLM(when comparing tables in the appendix produced by Qwen versus those in the main paragraph), even though the trends are similar. This is my main concern for the paper. Especially for a benchmark, if the score or model ranking can change depending on which VLM/LLM backend is used for scoring, it becomes difficult to trust this measure of progress. This subjectivity potentially suggests that the evaluation protocol may need further stabilization.
> >
> > Thank you for raising this concern. Our would like to highlight that StructScore produces *stable and consistent model rankings across a wide range of evaluators*, even though absolute numeric values naturally vary across LLMs. We address it from multiple angles:
> >
> > - StructScore is specifically designed to reduce LLM hallucination by decomposing evaluation into fine-grained Q&A pairs and using a two-stage process: (i) answer generation and (ii) answer verification, rather than relying on a single global judgment. This makes the evaluation substantially more objective and focused on factual correctness. For non-natural structured visuals, no current VLM achieves perfect perception, so some variation in *absolute scores* is expected and reflects differences in evaluator capability, not metric instability.
> > - We claim that **ranking consistency** is relatively more important than raw numeric values for a benchmark. Across both StructEditBench and StructT2IBench, the rankings produced by GPT-5 and Qwen2.5-VL-72B are fully consistent. Moreover, both evaluators exhibit strong alignment with human preferences (Pearson r > 0.9), as shown in our human ELO study.
> > - To further validate metric stability, we conducted an additional experiment during the rebuttal. Using a 1,000-sample subset, we compared StructScore using five different evaluators: GPT-5, GPT-5-mini, Qwen2.5-VL-72B, Qwen3-VL-8B, and Qwen3-VL-30B-A3B. Despite large differences in model architecture (open/closed source, dense/MoE), the **model rankings remain nearly identical across all evaluators**. This indicates that StructScore is **stable**, and changes to the judging model do not lead to ranking flips.
> >
> > | | Qwen2.5-VL 72B | Qwen3-VL 30B-A3B | Qwen3-VL 32B | GPT-5-mini | GPT-5 |
> > |---|---|---|---|---|---|
> > | UniWorld-V1 | 12.12 (14) | 7.88 (14) | 7.76 (14) | 8.25 (14) | 7.37 (14) |
> > | DiMOO | 15.75 (13) | 19.11 (13) | 18.90 (13) | 21.23 (13) | 20.09 (13) |
> > | OmniGen2 | 19.88 (12) | 23.18 (12) | 21.54 (12) | 22.46 (12) | 23.35 (12) |
> > | Ovis-U1 | 21.12 (11) | 23.84 (11) | 23.71 (11) | 27.02 (11) | 27.08 (11) |
> > | Bagel | 26.48 (10) | 27.99 (10) | 27.99 (9) | 27.87 (10) | 27.80 (10) |
> > | HiDream-E1.1 | 25.25 (9) | 28.24 (9) | 27.27 (10) | 28.40 (9) | 28.56 (9) |
> > | Bagel-Think | 28.56 (8) | 29.96 (8) | 30.57 (8) | 31.20 (8) | 32.25 (8) |
> > | Step1X-Edit | 29.85 (7) | 31.68 (7) | 32.13 (7) | 33.73 (7) | 33.18 (7) |
> > | FLUX.1 Kontext | 33.05 (6) | 34.56 (6) | 35.80 (5) | 37.80 (6) | 36.58 (6) |
> > | Qwen-Edit | 33.52 (5) | 35.68 (5) | 35.50 (6) | 38.12 (5) | 37.16 (5) |
> > | Nano Banana | 46.18 (3) | 46.84 (4) | 48.92 (4) | 50.19 (4) | 50.47 (4) |
> > | GPT-Image | 46.01 (4) | 46.91 (3) | 49.25 (3) | 51.69 (3) | 51.29 (3) |
> > | Seedream 4.0 | 46.55 (2) | 47.04 (2) | 50.59 (2) | 52.73 (2) | 51.92 (2) |
> > | Ours | 48.26 (1) | 49.77 (1) | 51.49 (1) | 53.61 (1) | 54.92 (1) |

---

### Official Review · Reviewer_ZnGr · 2025-10-31

**Soundness:** 4
**Presentation:** 4
**Contribution:** 4
**Rating:** 8
**Confidence:** 3

**Summary:**

The paper proposes a large-scale dataset tackling the structured visuals editing problem, along with a novel benchmark, StructBench, and metric StructScore to evaluate the quality of edits. The proposed dataset, with the three-stage training paradigm introduced, helps achieve state-of-the-art performance in the StructBench benchmark.

**Strengths:**

1. The proposed large-scale dataset contains image pairs and respective drawing programs and editing instructions, which are useful for future advancements in text-to-structured visual generation or editing. With the drawing program info, it can also be beneficial to text-to-vector-based (such as SVG) diagram generation by simply changing the output format for image rendering.

2. The proposed StructBench, along with the novel metric, StructScore, is superior to traditional editing benchmarks with metrics such as PSNR, as demonstrated by the high correlation with human preference. The addition of Qwen-based evaluator also rules out the reproducibility concerns of closed-source models such as GPT-5.

3. The three-stage training approach in the paper effectively bridges Qwen2.5-VL with FLUX.1, achieving state-of-the-art performance on the proposed benchmark, while maintaining competitive general editing capabilities.

**Weaknesses:**

- Some minor unclear points in the paper structure. In L365, there are mentions of StructEditBench and StructT2IBench. In section 3, only StructBench was introduced. I assume the Q&A construction and metrics around L244 are applied to both editing and T2I settings, but it should be clearly explained in this section.

**Questions:**

I am happy with the paper overall. Some technical details I would like to ask

- In Appendix A.2, only part of the training hyperparameters were provided. It would be helpful to include additional info such as GPU hours, number of iterations, LR scheduling, etc., given that the three-stage training involves fairly large datasets such as FLUX-Reason-6M and the one proposed in this paper.

---

> ### Author Response · Authors · 2025-11-21
>
> We deeply appreciate your insightful comments and efforts in reviewing our manuscript. We respond to each of your comments one-by-one in what follows. In the revised draft, we mark our major revisions as “red”.
>
> > Q1: Some minor unclear points in the paper structure. In L365, there are mentions of StructEditBench and StructT2IBench. In section 3, only StructBench was introduced. I assume the Q&A construction and metrics around L244 are applied to both editing and T2I settings, but it should be clearly explained in this section.
>
> Thank you for pointing this out. You are correct that the Q&A construction process and the StructScore metric described in Section 3 apply identically to both StructEditBench and StructT2IBench. This is because both tasks share the same ground-truth target image, allowing the same fine-grained Q&A pairs to evaluate factual correctness in either setting.
>
> We have clarified this explicitly in the revised paper by (i) explaining that StructBench is the overall name consisting of both editing and T2I subsets, and (ii) stating that the Q&A generation and evaluation protocol is uniformly applied to both. This should eliminate confusion and make the relationship between StructBench, StructEditBench, and StructT2IBench clearer.
>
> > Q2: In Appendix A.2, only part of the training hyperparameters were provided. It would be helpful to include additional info such as GPU hours, number of iterations, LR scheduling, etc., given that the three-stage training involves fairly large datasets such as FLUX-Reason-6M and the one proposed in this paper.
>
> Thank you for the suggestion. We agree that providing more comprehensive training details would improve clarity and reproducibility. In the revised version, we have added the missing information to Appendix A.2, including GPU hours, number of iterations for each training stage, and the learning-rate schedules. Note that we only use a small proportion of datasets like FLUX-Reason-6M since our aim is just to keep model capability on natural images from degradation. We hope these updates make the training pipeline clearer and easier to reproduce.
>
> |               | stage1 | stage2 | stage3 |
> |---------------|--------|--------|--------|
> | steps         | 16000  | 6000   | 8000   |
> | gpu time (h100 hours)      | 4480| 1600 | 2560 |
> | lr schedule   | constant with warmup      |

---

### Official Review · Reviewer_pUs6 · 2025-11-01

**Soundness:** 3
**Presentation:** 3
**Contribution:** 3
**Rating:** 6
**Confidence:** 3

**Summary:**

The paper targets structured image generation/editing (charts, math/graph diagrams, tables, scientific schematics), where current T2I/VL models often produce plausible but factually wrong images. The authors build a 1.3M code-aligned dataset with paired instructions and CoT, plug Qwen-VL into FLUX.1-Kontext via a lightweight connector, train in three stages, and propose StructBench + StructScore to evaluate fine-grained factuality. They report strong gains over open models and show that adding explicit reasoning helps other models too

**Strengths:**

- Well-motivated task: clearly shows why structured visuals are different from “pretty pictures” and why factuality/layout need a dedicated treatment.
- Clean data pipeline: executable graphics → instruction + code edit - re-render - filtered, so supervision is tight and verifiable.
- Useful benchmark: StructBench/StructScore gives a more faithful measure than CLIP-style metrics and correlates with human judgment.
- Clarity and Comprehensiveness: The paper is exceptionally well-written and structured. The problem, contributions, and methodology are articulated with clarity.

**Weaknesses:**

- Dataset Domain and Generalizability: The reliance on executable programs for data generation may introduce a significant domain gap. It is questionable whether this dataset fully captures the diversity, noise, and "messiness" of structured visuals found in real-world sources

**Questions:**

- Have you considered evaluator bias? In StructScore, if the VLM used to ask questions and the VLM used to answer them belong to the same model family as the model being evaluated, how do you avoid evaluation bias or overfitting to the evaluator’s visual priors?

- Have you considered how stable the automatic metric is across different model versions? Could small changes in the judge cause the model rankings to change?

---

> ### Author Response · Authors · 2025-11-21
>
> We deeply appreciate your insightful comments and efforts in reviewing our manuscript. We respond to each of your comments one-by-one in what follows. In the revised draft, we mark our major revisions as “red”.
>
> > Q1: Dataset Domain and Generalizability: The reliance on executable programs for data generation may introduce a significant domain gap. It is questionable whether this dataset fully captures the diversity, noise, and "messiness" of structured visuals found in real-world sources
>
> Thank you for raising this important point. We agree that synthesized data constructed from executable programs cannot fully match the diversity, noise, and “messiness” of real-world structured visuals. We address this concern when designing our framework in three parts:
>
> - First, we have intentionally collected data from a wide range of sub-domains, such as geometry, graphs, documents, scientific figures, charts, tables, to maximize coverage. **Because factual accuracy is especially critical for structured images, we deliberately prioritize reliability and precise code–image alignment over maximal visual diversity when designing our data pipeline.**
> - Second, as this is the first systematic study focused on *structured* image generation and editing, we find that even leading models (e.g., GPT-Image, Nano Banana) struggle on what appear to be relatively simple synthetic cases (accuracy < 50% in Tables 1, 2, and 3). This indicates that **current models are far from saturating the current benchmark.** Our goal is therefore to take a staged approach—similar to how MMMU [1] evolved into MMMU-Pro [2], where more sophisticated visual augmentations and real-world noise can be introduced *after* models demonstrate strong performance on the clean, verifiable version of the task. For example, evaluating aesthetics or real-world presentation variability is indeed a valuable direction, but one that becomes meaningful only after models can reliably handle the foundational structured content.
> - Finally, to showcase the generalizability, **we use the latest Nano Banana 2.0 to augment our benchmark data to simulate “real-world” images**, e.g., transforming simple white background into complex scene like white board and computer screen. As shown in Fig. 14 in the  revised draft, our model, while only trained on code-rendered images, showcases the generalization capability to edit real-world images with a significant domain gap.
>
>
> **References**
>
> [1] Yue, Xiang, et al. "Mmmu: A massive multi-discipline multimodal understanding and reasoning benchmark for expert agi." *Proceedings of the IEEE/CVF Conference on Computer Vision and Pattern Recognition*. 2024.
>
> [2] Yue, Xiang, et al. "Mmmu-pro: A more robust multi-discipline multimodal understanding benchmark." *Proceedings of the 63rd Annual Meeting of the Association for Computational Linguistics (Volume 1: Long Papers)*. 2025.

---

> > ### Author Response · Authors · 2025-11-21
> >
> > > Q2: Have you considered evaluator bias? In StructScore, if the VLM used to ask questions and the VLM used to answer them belong to the same model family as the model being evaluated, how do you avoid evaluation bias or overfitting to the evaluator’s visual priors?
> > > Q3: Have you considered how stable the automatic metric is across different model versions? Could small changes in the judge cause the model rankings to change?
> >
> > Thank you for raising these two closely related questions about evaluator bias and metric stability. We address both concerns below.
> >
> > First, StructScore is intentionally designed *not* to use a single VLM for end-to-end scoring, which could introduce subjective bias. Instead, **we explicitly decompose evaluation into two stages**:
> >
> > (1) generating fine-grained Q&A pairs that target factual attributes, and
> >
> > (2) verifying answers through an additional comparison step
> >
> > This separation reduces reliance on a single model’s visual priors and makes the evaluation focus on factual correctness rather than stylistic similarity.
> >
> > Second, we provide results under **two independent evaluators** in the paper, i.e., GPT-5 and Qwen2.5-VL-72B; and we show in Figures 6 and 10 that both of them strongly correlate with human preference judgments.
> >
> > To more directly address the concern about metric stability across model versions, **we conducted an additional experiment as shown in the following table.** We sampled 1,000 examples from StructEditBench and evaluated five different VLMs as judges: GPT-5-mini, GPT-5, Qwen2.5-VL-72B, Qwen3-VL-Dense-32B, and Qwen3-VL-30B-A3B. While absolute scores vary due to differences in multimodal capability, **the overall ranking of evaluated models remains highly consistent across all evaluators,** including closed-source vs. open-source and dense vs. MoE architectures. This indicates that StructScore is stable, and small changes to the judging model do not lead to ranking flips.
> >
> >
> > | | Qwen2.5-VL 72B | Qwen3-VL 30B-A3B | Qwen3-VL 32B | GPT-5-mini | GPT-5 |
> > |---|---|---|---|---|---|
> > | UniWorld-V1 | 12.12 (14) | 7.88 (14) | 7.76 (14) | 8.25 (14) | 7.37 (14) |
> > | DiMOO | 15.75 (13) | 19.11 (13) | 18.90 (13) | 21.23 (13) | 20.09 (13) |
> > | OmniGen2 | 19.88 (12) | 23.18 (12) | 21.54 (12) | 22.46 (12) | 23.35 (12) |
> > | Ovis-U1 | 21.12 (11) | 23.84 (11) | 23.71 (11) | 27.02 (11) | 27.08 (11) |
> > | Bagel | 26.48 (10) | 27.99 (10) | 27.99 (9) | 27.87 (10) | 27.80 (10) |
> > | HiDream-E1.1 | 25.25 (9) | 28.24 (9) | 27.27 (10) | 28.40 (9) | 28.56 (9) |
> > | Bagel-Think | 28.56 (8) | 29.96 (8) | 30.57 (8) | 31.20 (8) | 32.25 (8) |
> > | Step1X-Edit | 29.85 (7) | 31.68 (7) | 32.13 (7) | 33.73 (7) | 33.18 (7) |
> > | FLUX.1 Kontext | 33.05 (6) | 34.56 (6) | 35.80 (5) | 37.80 (6) | 36.58 (6) |
> > | Qwen-Edit | 33.52 (5) | 35.68 (5) | 35.50 (6) | 38.12 (5) | 37.16 (5) |
> > | Nano Banana | 46.18 (3) | 46.84 (4) | 48.92 (4) | 50.19 (4) | 50.47 (4) |
> > | GPT-Image | 46.01 (4) | 46.91 (3) | 49.25 (3) | 51.69 (3) | 51.29 (3) |
> > | Seedream 4.0 | 46.55 (2) | 47.04 (2) | 50.59 (2) | 52.73 (2) | 51.92 (2) |
> > | Ours | 48.26 (1) | 49.77 (1) | 51.49 (1) | 53.61 (1) | 54.92 (1) |

---

> > > ### Comment · Reviewer_pUs6 · 2025-11-27
> > >
> > > Thank you for your response; this has resolved my concerns.

---

> > > > ### Author Response · Authors · 2025-11-27
> > > >
> > > > Thanks for your feedback and wishing you a joyful Thanksgiving!

---

### Official Review · Reviewer_HadZ · 2025-11-05

**Soundness:** 3
**Presentation:** 3
**Contribution:** 3
**Rating:** 6
**Confidence:** 3

**Summary:**

This paper tackles a point in current image generation: models are great at making artistic "vibes" but absolutely terrible when you need them to be factually accurate, like drawing a specific chart, a math diagram, or a table.
To fix this, the authors didn't scrape the web; they generated a huge dataset (1.3M pairs) by running actual code (like Python/LaTeX), ensuring perfect ground truth. They then trained a FLUX-based model using a three-stage curriculum designed to make the model "think" more about structure before generating. They also had StructBench, a benchmark for these tasks. Seems like this benchmark is difficult.

**Strengths:**

Tackles an under-researched area where standard aesthetic-focused models fail precisely because high factual accuracy (text rendering, exact layout) is required.


Good idea: Utilizing programmatically generated images (via code) creates a verifiable and noise-free ground truth, superior to scraping diverse but unreliable web data for this specific task.

StructScore improves upon naive "VLM-as-a-judge" approaches by using fine-grained, verified Q&A pairs, showing strong alignment with human evaluators.

Integration of Reasoning in the paper

**Weaknesses:**

1. Do we really need image editing to do this task? People can just ask LLM to write code to generate the new image, which is more accurate. With pixel level image generation model, there will be inevitable artifacts.

2. How to make sure this data curation pipeline is accurate? Any error rate statistics?

3. No evaluation in addition to infographics. You should evaluate on the natural image editing task.

**Questions:**

See above comments.

---

> ### Author Response · Authors · 2025-11-21
>
> We deeply appreciate your insightful comments and efforts in reviewing our manuscript. We respond to each of your comments one-by-one in what follows. In the revised draft, we mark our major revisions as “red”.
>
> > Q1: Do we really need image editing to do this task? People can just ask LLM to write code to generate the new image, which is more accurate. With pixel level image generation model, there will be inevitable artifacts.
>
> Using LLMs to write code for image generation or editing can indeed achieve good results. However, we believe that modeling such structured images is still very important for visual generation models, for several reasons:
>
> - **First, we believe the shift from aesthetics-driven generation toward factuality-driven generation is a natural evolution of visual generation systems.** As mentioned in our paper, current visual generation models often over-optimize for high aesthetic quality, leading to the “AI look.” The recent Seedream 4.0 model by ByteDance [1] also points out this imbalance and constructs a large amount of non-natural structured images using LaTeX, OCR, etc. Similarly, the Nano Banana team claimed that factuality in future image generation/editing is crucial for supporting real-life applications (e.g., office scenarios). Their recent Nano Banana 2.0 demo further confirms this viewpoint. These observations align with our paper.
> - Second, for unified models, a major reason behind the generation–understanding gap is the large domain discrepancy. If a model is unable to perform even the most basic generation and editing tasks in these non-natural domains (i.e., there is a large mismatch between understanding and generation abilities), then it becomes difficult to achieve mutual between understanding and generation.
> - **Third, vision-based image editing aligns better with human perception.** Regular users (non-programmers) do not modify or create charts by writing code first and then rendering an image. Instead, they adjust visual elements directly on a draft, or within UI tools like Excel or PowerPoint. This inspires us: a unified model can adopt a similar approach where users specify modifications to visual elements directly. In contrast, an LLM must first convert a chart into code, reason about how to modify it in code space, generate the modified code, and then re-render the entire image. This multi-step process inevitably introduces additional errors. For example, if the editing instruction is “move the legend so it does not overlap the curve”, LLM-based method struggles to figure out how to modify the legend position since it has no idea of the curve’s exact shape. In contrast, this will be much eaiser to move the legend to an empty area in the visual domain.
> - Finally, as shown in Fig. 14 in the revised paper, **LLM-based methods cannot handle real-world images**, e.g., photo of chart on a whiteboard or geometric graph displayed on computer screen.
>
> > Q2: How to make sure this data curation pipeline is accurate? Any error rate statistics?
>
> - **All images and text instructions in our dataset are strictly aligned with the underlying program code to ensure accuracy.** As shown in Fig. 2, we generate editing instructions at both the image and code levels which improves the consistency. Moreover, we use a powerful LLM, GPT-5, throughout the entire pipeline to ensure the accuracy of the automated process. In contrast, traditional image-editing datasets are often synthesized by expert models [2,3], which inevitably introduces noise.
> - To further guarantee the correctness of the data construction pipeline, we apply strict filtering at every stage. For the training dataset, we use rule-based filters to remove code that fails to execute (≈1%) and image pairs where the visual differences are too small (≈0.5%). For the benchmark, we apply both GPT-5 and manual inspection to curate a high-quality test set.
>
> **References**
>
> [1] Seedream, Team, et al. "Seedream 4.0: Toward next-generation multimodal image generation." *arXiv preprint arXiv:2509.20427* (2025).
>
>
> [2] Brooks, Tim, Aleksander Holynski, and Alexei A. Efros. "Instructpix2pix: Learning to follow image editing instructions." *Proceedings of the IEEE/CVF conference on computer vision and pattern recognition*. 2023.
>
> [3] Wei, Cong, et al. "Omniedit: Building image editing generalist models through specialist supervision." The Thirteenth International Conference on Learning Representations. 2024.

---

> ### Author Response · Authors · 2025-11-21
>
> > Q3: No evaluation in addition to infographics. You should evaluate on the natural image editing task.
> - Our primary focus in this work is to investigate the capability boundary of *non-natural* structured visuals, so the main paper focuses on structured-image generation and editing. Nevertheless, we fully agree that a unified model should not sacrifice performance on natural-image editing. This is why we intentionally include natural-image data during training to preserve general-domain capability.
> - As shown in Table 8 of the Appendix, **we provide an evaluation of our model on the natural-image editing benchmark ImgEdit [1].** Our model not only avoids degradation on natural images but also improves over the FLUX-Kontext [2] baseline (3.75 vs. 3.52), demonstrating that unifying natural and non-natural image modeling is a feasible and promising direction. We have also added more qualitative demos of natural image editing in Fig. 10 of the revised version.
>
> **References**
>
> [1] Ye, Yang, et al. "ImgEdit: A Unified Image Editing Dataset and Benchmark." arXiv preprint arXiv:2505.20275 (2025).
>
> [2] Black Forest Labs, et al. "FLUX.1 Kontext: Flow Matching for In-Context Image Generation and Editing in Latent Space." arXiv preprint arXiv:2506.15742 (2025).

---

### Author Response · Authors · 2025-11-27

Dear Reviewers

We thank all the reviewers for the valuable feedback. Since the discussion phase is nearing completion, we wanted to follow up and ensure all your concerns have been fully addressed. If there are any further points or clarifications you would like from us, please feel free to let us know—we would be grateful for any additional feedback.

Thank you for your efforts in reviewing our work.

---

### Author Response · Authors · 2025-12-01

We sincerely thank the reviewers and chairs for their time and constructive feedback on our submission.

Our work focuses on an important yet under-explored problem in visual generation: generation and editing of *non-natural* (structured) images such as charts, diagrams, and tables, which reviewer HadZ, pUs6 found well motivated. Building on a code-aligned, scalable data pipeline, we construct a dataset of 1.3M high-quality examples with detailed chain-of-thought reasoning, which all four reviewers highlighted as a robust, controlled, clean method. We further introduce a corresponding benchmark and an evaluation metric whose scores align closely with human preferences; all reviewers considered this a useful benchmark (pUs6) with a novel metric (ZnGr). Finally, we propose a progressive training curriculum that yields a unified model with strong performance on both natural and non-natural images, which reviewer ZnGr found effective.

Rebuttal updates and manuscript changes:

* **More evaluator results.** We expanded StructScore results to include both open- and closed-source evaluators, dense and MoE models, and multiple model sizes. Across these settings, model rankings remain highly consistent (reviewer pUs6, vk2E).
* **More qualitative demos.** We added natural-image editing examples in Fig. 10 (reviewer HadZ) and show generalization to Nano Banana 2.0–augmented “real-world” structured images in Fig. 14 (reviewer pUs6).
* **Additional analysis and clarifications.** We further highlight the role of non-natural images for visual generation and unified models (reviewer HadZ), discuss the generalizability of synthesized data (reviewer pUs6), and provide extra training and experimental details  (reviewer ZnGr).

---

### Meta-Review · Area_Chair_S6qh · 2025-12-12

**Summary:**

This paper focuses on the limitation of image generation models in domains that require accuracy and factual fidelity, such as charts, graphs, tables. To address the issue, a large-scale dataset with 1.3M samples are constructed. To ensure accuracy, images are generated with executable code, and pairs are annotated by GPT-5. A model is then trained on the proposed dataset, and the trained model shows promising results on the proposed `StructBench`.

**Reviewer Concerns:**

Reviewers' concerns are mainly on the data curation pipeline (e.g., how to ensure accuracy and diversity. The AC has read the review and rebuttal and confirm that the concerns are mostly resolved by the authors with evidence.

**Reviewer Scores:**

The original scores are (6, 6, 6, 8), and there are no responses from the reviewer. Therefore the AC believes that they will maintain a similar score.

---

### Decision · Program_Chairs · 2026-01-26

Accept (Poster)